# Evidence for a deep, distributed and dynamic code for animacy in human ventral anterior temporal cortex

Timothy T Rogers[1]*, Christopher R Cox[2], Qihong Lu[3], Akihiro Shimotake[4], Takayuki Kikuchi[5], Takeharu Kunieda[5,6], Susumu Miyamoto[5], Ryosuke Takahashi[4], Akio Ikeda[7], Riki Matsumoto[4,8], Matthew A Lambon Ralph[9]*

[1]Department of Psychology, University of Wisconsin- Madison, Madison, United States; [2]Department of Psychology, Louisiana State University, Baton Rouge, United States; [3]Department of Psychology, Princeton University, Princeton, United States; [4]Department of Neurology, Kyoto University Graduate School of Medicine, Kyoto, Japan; [5]Department of Neurosurgery, Kyoto University Graduate School of Medicine, Kyoto, Japan; [6]Department of Neurosurgery, Ehime University Graduate School of Medicine, Ehime, Japan; [7]Department of Epilepsy, Movement Disorders and Physiology, Kyoto University Graduate School ofMedicine, Kyoto, Japan; [8]Division of Neurology, Kobe University Graduate School of Medicine, Kusunoki-cho, Kobe, Japan; [9]MRC Cognition and Brain Sciences Unit, University of Cambridge, Cambridge, United Kingdom

**Abstract** How does the human brain encode semantic information about objects? This paper reconciles two seemingly contradictory views. The first proposes that local neural populations independently encode semantic features; the second, that semantic representations arise as a dynamic distributed code that changes radically with stimulus processing. Combining simulations with a well-known neural network model of semantic memory, multivariate pattern classification, and human electrocorticography, we find that both views are partially correct: information about the animacy of a depicted stimulus is distributed across ventral temporal cortex in a dynamic code possessing feature-like elements posteriorly but with elements that change rapidly and nonlinearly in anterior regions. This pattern is consistent with the view that anterior temporal lobes serve as a deep cross-modal 'hub' in an interactive semantic network, and more generally suggests that tertiary association cortices may adopt dynamic distributed codes difficult to detect with common brain imaging methods.

*For correspondence:
ttrogers@wisc.edu (TTR);
Matt.Lambon-Ralph@mrc-cbu.
cam.ac.uk (MALR)

## Introduction

Semantic memory supports the remarkable human ability to recognize new items and events, infer their unobserved properties, and comprehend and produce statements about them (*McClelland and Rogers, 2003*; *Rogers and McClelland, 2004*). These abilities arise from neural activity propagating in a broadly distributed cortical network, with different components encoding different varieties of information (perceptual, motor, linguistic, etc *Martin, 2016*; *Huth et al., 2016*; *Patterson et al., 2007*). The ventral anterior temporal lobes (vATL) form a deep hub in this network that coordinates activation amongst the various surface representations (*Patterson et al., 2007*; *Lambon Ralph et al., 2017*). In so doing, the vATL acquires distributed representations that allow the whole network to express conceptual similarity structure, supporting inductive generalization of acquired knowledge across conceptually related items (*Rogers et al., 2004*).

Within this framework, contemporary theories differ in their proposals about the semantic representation of visually-presented objects. Many researchers view such representations as arising from the propagation of activity upward through a feature hierarchy, beginning with simple perceptual features, moving through intermediate modality-specific features, and ending with abstract cross-modal features that express semantic category structure. This perspective has strongly influenced computational models of object recognition developed in visual neuroscience (*Isik et al., 2014*; *Serre et al., 2007*) and associated brain imaging work (*Cichy et al., 2016*; *Kriegeskorte, 2015*). Alternatively, semantic representations may arise from interactive settling processes within distributed and dynamic brain networks (*Patterson et al., 2007*; *Chen et al., 2017*). On this view, semantic representations are activation patterns encoded jointly over many neural populations, with similar concepts evoking similar patterns. The information encoded by a given population depends on the states of other populations, so that the whole representation is best viewed as a point in a multidimensional state space (*Rogers and McClelland, 2004*). Stimulus processing involves a transitioning of the system through the space, rather than the activation of increasingly abstract/complex feature detectors, as the whole system settles toward an interpretation of the input (*Rogers and Patterson, 2007*). Many models of semantic memory adopt this perspective (*Chen et al., 2017*; *Farah and McClelland, 1991*; *Cree et al., 1999*; *Harm and Seidenberg, 2004*; *Plaut and Shallice, 1993*), which also aligns with the general view that the ventral visual processing stream is recurrent and interactive (*Kravitz et al., 2013*; *Goddard et al., 2016*; *Kietzmann et al., 2019*).

The two views carry critically different implications for understanding how neural systems represent semantic information. On featured-based views, local populations of neurons independently encode the presence of a particular feature, licensing a straightforward interpretation of neural activity: when the population is active, the corresponding feature has been detected, inferred, or called to mind; when inactive, it hasn't. The temporal behavior of the population directly indicates the time-course

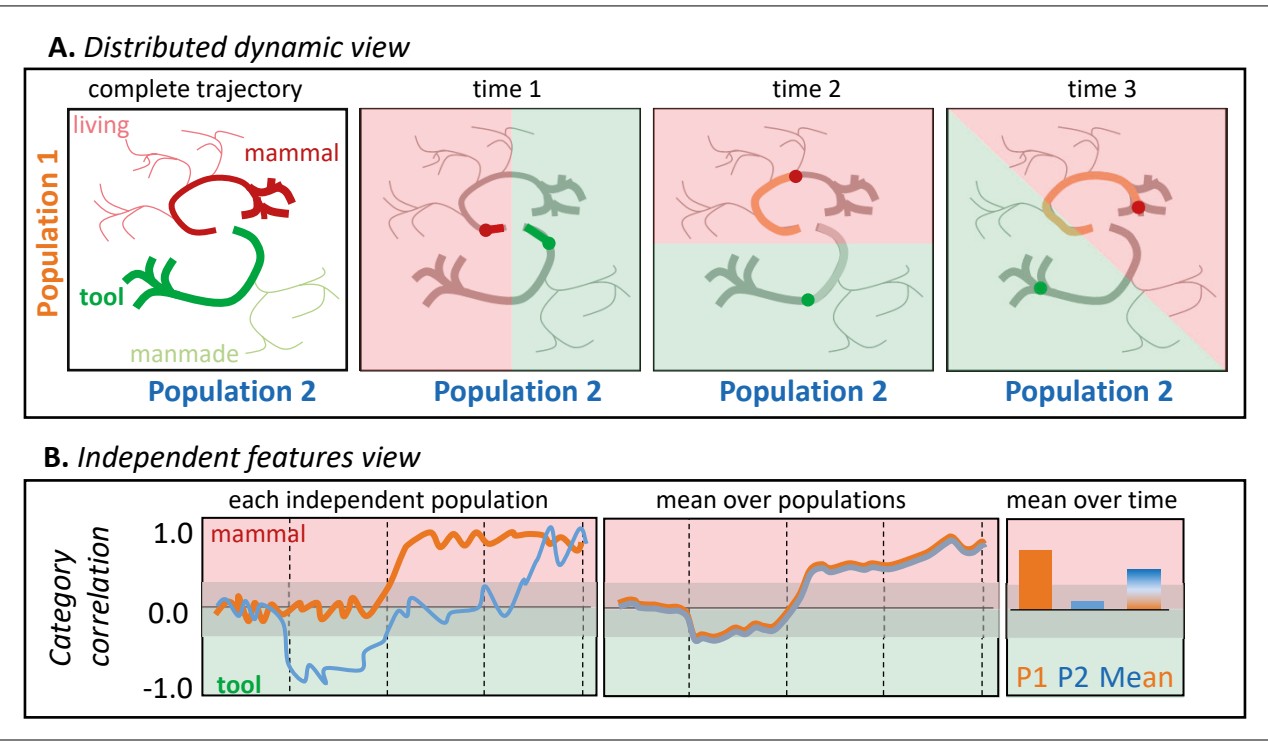

**Figure 1.** Two views of neural representation. A. Hypothetical joint activations of two neural populations to living and manmade items (left), and the classification plane that would best discriminate tools from mammals at different timepoints. Jointly the two populations always discriminate the categories, but the contribution of each population to classification changes over time so that the classification plane rotates. B. Independent correlations between each population's activity and a binary category label (tool/mammal) for the same trajectories plotted above, shown across time for each population (left), averaged across the two populations (middle), and averaged over time for each population independently or for both populations (right). Independent correlations suggest conclusions about when and how semantic information is represented that are incorrect under the distributed and dynamic view.

with which the represented feature is available to influence processing, and the mean neural activity over time indicates the strength with which the feature was activated in a particular trial or task condition. These ideas motivate efforts to determine which cortical regions encode which features at which points in time by correlating/predicting the presence of a semantic feature (such as a category label or property) with/from local neural activity (*Martin, 2016*; *Pulvermüller, 2013*). If, however, semantic representations are distributed, the local behavior of a single population may not be interpretable independent of the others (*Cox et al., 2014*); and if semantic processing is also dynamic, the contribution of a local population to the distributed code may change in real time, potentially in highly nonlinear ways (*Mante et al., 2013*). These possibilities suggest that the treatment of neurophysiological signals as arising from independent feature detectors can mischaracterize how neural systems encode semantic information.

*Figure 1* illustrates why. Suppose a participant names images of tools and mammals while the responses of two neural populations are measured with high temporal and spatial resolution. The top panels show hypothetical joint responses to each stimulus over time from onset to response (together with those of other untested living and manmade items), plotted as trajectories in a 2D space over time. The shaded panels show how a multivariate classifier partitions the space when trained to discriminate mammals from tools at one timepoint. The categories are always well differentiated, so the two populations jointly encode semantic information at every timepoint, but because the trajectories are nonlinear each population's contribution to this structure changes over time—the category discrimination plane rotates. The bottom panels show how each population's behavior would appear if its activity was analyzed independently via correlation with semantic information (e.g. by computing the Pearson correlation between the population activity for each stimulus at one point in time and a binary semantic category label). The orange line shows how Population 1's activity (i.e. the projection of the trajectories in panel A against the vertical axis only) correlates with the tool/mammal label at each timepoint—the correlation becomes positive about half-way through the time series. The blue line shows how Population 2's activity (i.e. the Panel A trajectories projected onto the horizontal axis) correlates with the tool/mammal label: it quickly shows a negative correlation (tools more active than mammal), then no correlation, then a positive correlation (mammals more active than tools). From these independent analyses, it might seem that (**a**) animals are detected only midway through the time-series, (**b**) tools are detected earlier than animals, and (**c**) there exist populations that 'switch' from tool-detectors to mammal-detectors. The bottom middle panel shows the mean response across the two populations at each point in time, as might be observed if the neurophysiological measurement lacks spatial resolution (e.g. EEG/MEG). The populations appear to detect animals late in the trial. The right panel shows mean responses across time for each population (consistent with high spatial and low temporal resolution; e.g. fMRI) and for both populations together (low spatial and temporal resolution). The former suggests that population two plays no important role distinguishing the categories, while the latter suggests the two populations together selectively detect animals. These conclusions are incorrect from the distributed and dynamic perspective, under which the two populations always jointly differentiate the categories but the contribution of each changes over time (*Figure 1A*).

The interpretation of neurophysiological signals in the cortical semantic system thus depends critically upon whether the neuro-semantic code is feature-based or distributed and dynamic at whatever scale measurements are taken. Efforts to adjudicate the question face three significant hurdles. First, spatial and/or temporal averaging can obscure signal if the code truly is distributed and dynamic—discovery requires neural data with high temporal and spatial resolution, ruling out the non-invasive methodologies that constitute the majority of work in this area (*Contini et al., 2017*). Second, independent univariate analysis can mischaracterize information distributed across multiple channels—discovery requires multivariate methods (*Grootswagers et al., 2017*). Third, studies connecting neurophysiological measurements to computational models of visual recognition have primarily focused on feed-forward models that do not exhibit dynamic processing (*Cichy et al., 2016*; *Clarke et al., 2015*). While some recent decoding evidence suggests visual processing in ventral temporal cortex is dynamic (*Kietzmann et al., 2019*), no prior work has assessed how representations evolve within distributed and dynamic semantic models, and consequently it is unclear what this view predicts or how it differs from feature-based approaches.

We therefore combined computational modeling, multivariate pattern classification, and electro-corticography (ECoG) to assess whether semantic representations in human vATL are distributed and dynamic. We focused on vATL due to its central role in semantic representation (*Patterson et al., 2007*; *Lambon Ralph et al., 2017*) and because it is a common target for grid implantation in patients awaiting surgery to remediate epilepsy. We first showed that the nonlinear dynamics highlighted in our thought experiment arise in a well-studied neurocomputational model of semantic memory (*Rogers et al., 2004*; *Chen et al., 2017*), then used temporal generalization of pattern classifiers (*King and Dehaene, 2014*) to establish the predicted 'signature' of a dynamically changing semantic code (is the stimulus animate or inanimate?) under this model. We next applied this technique to intracranial local field potentials collected from the surface of the left vATL while participants named line drawings of common items, and found the critical decoding signature—providing strong evidence that, at the level of spatiotemporal resolution captured by ECoG, the animacy of a visually depicted item is expressed in vATL by a distributed code that changes dynamically with stimulus processing. Finally we considered how the neural code exploited by classifiers changed over time and across the ventral temporal surface, with results that reconcile the two divergent views of semantic representation in the brain.

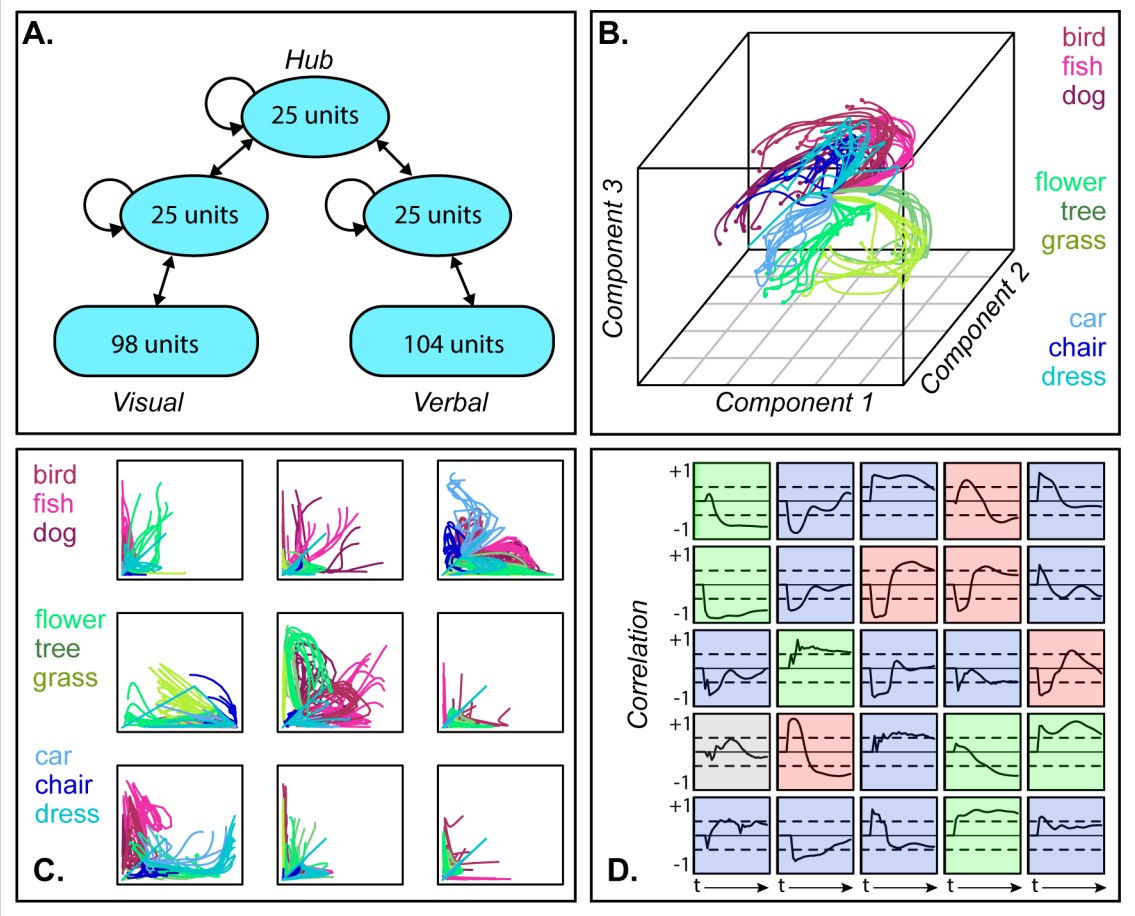

**Figure 2.** Dynamic representation in a neural network model of semantic processing. A. Model architecture. B. 3D MDS of hub activation patterns learned in one model run—each line shows the trajectory of a single item over time in the compressed space. C. The same trajectories shown in uncompressed unit activations for nine randomly sampled unit pairs, horizontal and vertical axes each showing activation of one unit. D. Feature-based analysis of each hub unit in one network run. Each square shows one unit. Lines trace, across time, the correlation between unit activation and category labels across items with dashed lines showing significance thresholds. Color indicates different patterns of responding (see text).

# Results

## Simulation study

Simulations provide a formal basis for understanding the implications of the distributed and dynamic view, because a computer model's architecture, behavior, learning, and testing patterns are fully known. We therefore adapted the well-known 'hub and spokes' model of semantic memory (*Patterson et al., 2007*; *Chen et al., 2017*; *Rogers et al., 2004*) to assess whether semantic representations change dynamically with stimulus processing and how the temporal generalization method can uncover such a code.

The model was a fully continuous and recurrent neural network that learns cross-modal associations between distributed visual and verbal representations via three reciprocally connected hidden layers (*Figure 2A*). Given positive input to a subset of visual or verbal units, it learns to activate the corresponding item's unspecified visual and/or verbal attributes. Activity propagates in both directions, from surface representations to hub and back, so that the trained model settles to an attractor state representing any item specified by the input. We trained the model to complete patterns representing ninety items from three model conceptual domains (e.g. animals, objects, and plants), each organized into three categories containing ten items (see Materials and methods for details). Items from different categories in the same domain shared a few properties while those in the same category shared many. We then presented the trained model with visual input for each item and recorded the resulting activation patterns over hub units for each tick of simulated time as the network settled.

To visualize how model internal representations changed during stimulus processing, we computed a 3D multi-dimensional scaling (MDS) of activation patterns at all timepoints during 'perception' of each stimulus, then plotted the changing representation for each item as a line in this space. The result in *Figure 2B* shows a systematic but nonlinear elaboration of the conceptual structure latent in the stimulus patterns: the domains separate from one another early on, but each item follows a curved trajectory over the course of stimulus processing. The curves are not an artifact of data-reduction—*Figure 2C* shows these trajectories in the native activations of randomly-sampled unit pairs in one network run, where they are clearly nonlinear (and see A-1 for comparative results with linearly-changing representations). Consequently, independent correlation analysis of each unit's behavior produces mixed results (*Figure 2D*), with some behaving like tonic category detectors (green squares), some like transient detectors (blue), some appearing to flip their category preference (red) and others appearing not to code category information at all (gray).

Thus, the full distributed pattern across hub units elaborates conceptual structure from early in processing, but the progression is nonlinear and only clearly discernable in a low-dimensional embedding of the space. Such an embedding can be computed for the model because we know which units are important and can apply the MDS to all and only those units. The same approach cannot be applied to ECoG data for two reasons. First, one cannot know a priori which channels record signals relevant for semantic representation and thus cannot simply compute a low-dimensional embedding of all data collected. Instead one must fit a statistical model that will selectively weight signals useful for discerning semantic structure. Second, whereas the model allows access to the entire network, a cortical surface sensor array only sparsely samples the full range of field potentials arising from stimulus processing. The problem thus requires a multivariate statistical model capable of revealing a dynamically changing neural code when fitted to sparsely sampled neural data.

We therefore used multivariate logistic classifiers to decode semantic category information from hub activation patterns and assessed their behavior on simulated ECoG data, taking unit activation as a model proxy for the potential measured at a subdural electrode. For each simulated participant, we selected a sparse random subsample (15%) of all hub units—analogous to the sparse sampling of field potentials provided by a cortical sensor array—and recorded their responses to each stimulus at every tick of time. We fitted a separate classifier at each timepoint to distinguish two semantic domains from the activation patterns elicited over the subsampled units, analogous to the binary decoding of broad semantic property such as animacy. *Figure 3A* shows the cross-validated accuracy at each timepoint averaged over many network runs and subsamples. The classifiers performed well above chance as soon as input activation reached the hub units and throughout the time window.

To assess representational change over time, we next adopted a temporal generalization approach (*King and Dehaene, 2014*), using the classifier fitted at one timepoint to decode the patterns observed at each other timepoint. Accuracy should remain high if the information a classifier exploits

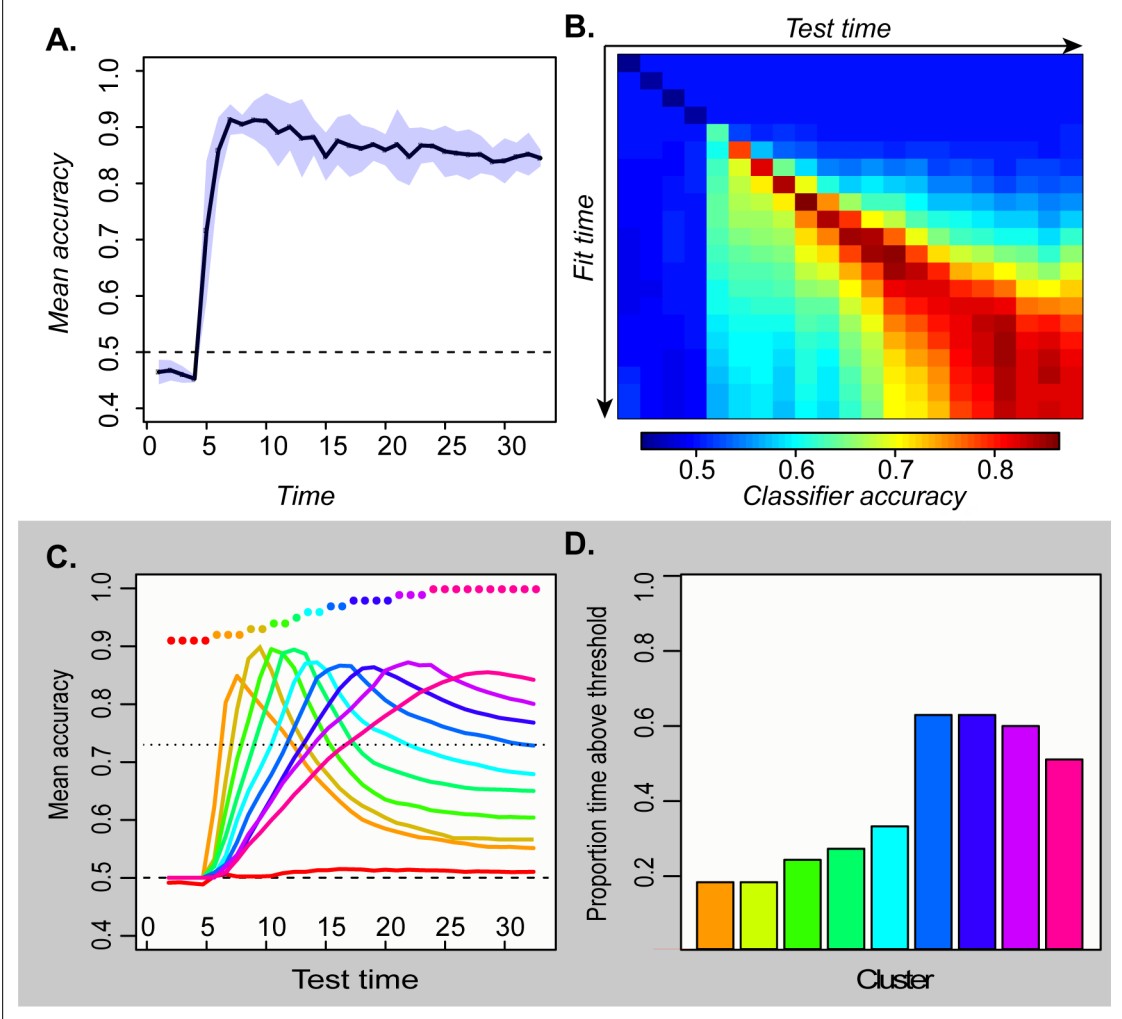

**Figure 3.** Temporal generalization profiles for deep network. A. Mean and 95 % confidence interval of the hold-out accuracy for classifiers trained at each tick of time in the model. B. Accuracy for each classifier (rows) tested at each point in time (columns). C. Mean accuracy for each cluster of classifiers at every point in time. Colored dots show the timepoints grouped together in each cluster. D. Proportion of the full time-window for which mean classifier accuracy in each cluster was reliably above chance.

at the training time persists at other timepoints. The temporal generalization profile of each classi-fier thus indicates how the underlying neural code persists or changes over time. Classifiers fitted to earlier activation patterns generalized only to immediate temporal neighbors, while those fitted to later patterns generalized over a wider window but failed at decoding earlier states (*Figure 3B*). To better visualize these results, we clustered the rows of the matrix in *Figure 3B* and plotted the mean accuracy of the classifiers in each cluster across time (*Figure 3C*). The results exhibit an 'overlapping waves' pattern: classifiers that work on early patterns quickly fail at later timepoints where a different classifier succeeds. As time progresses, the clusters include more classifiers and the breadth of time over which the classifiers perform well widens (*Figure 3D*).

This pattern reflects the nonlinear trajectories apparent in the sparsely-sampled representational space as the network settles to an attractor state. When trajectories curve, earlier classification planes fail later in time while later planes fail at earlier time-points. If representations simply moved linearly from initial to final state, early classifiers would continue to perform well throughout processing—a pattern observed in simulations with feature-based models, models with distributed representations that evolve linearly, and recurrent but shallow neural networks (see A-1). In the deep network, the non-linear dynamic pattern was observed only in the hub layer—in more superficial layers, the code remained stable (A-2). The simulations thus suggest that distributed and dynamic semantic represen-tations can arise in deep layers of interactive networks and a deep, distributed, and dynamic semantic

code of this kind will elicit a particular 'signature' when multivariate pattern classifiers are used to decode a binary semantic property such as animacy from ECoG data. Specifically, such classifiers will show:

1. *Constant decodability*. Neural activity predicts stimulus category at every time point once activation reaches the vATL.
2. *Local temporal generalization*. Classifiers generalize best to immediate temporal neighbors and worst to distal timepoints.
3. *Widening generalization window*. The temporal window over which classifiers generalize grows wider over time.
4. *Change in code direction*. Elevated neural activity can signify different semantic information at different points in time.

These are the characteristics we looked for in the ECoG study.

## ECoG study

The dataset included voltages collected at 1000 Hz from 16 to 24 subdural electrodes situated in the left ventral anterior temporal cortex of 8 patients awaiting surgery while they named line-drawings of common animate and inanimate items matched for a range of confounds (see Materials and methods). We analyzed voltages over the 1640ms following stimulus onset, a window that allows us to assess decodeability both before and after the mean onset of naming (1190 ms). Voltages were decoded using a 50 ms sliding-window, with separate classifiers fitted at each window and the window advancing in 10 ms increments. This yielded 160 classifiers per subject, each decoding potentials measured across all vATL electrodes in a 50 ms window. Each classifier was then tested on all 160 time-windows. The classifiers were logistic regression models trained to discriminate animate from inanimate items based on the voltages measured across all electrodes and for all time-points within the corresponding window. The models were fitted with L1 regularization to encourage coefficients of 0 for many features (see Materials and methods).

Hold-out accuracy exceeded chance at about 200 ms post stimulus onset, well before name initiation, and remained statistically reliable throughout the time window (*Figure 4A*) with no obvious change at the mean time of naming onset. At 200 ms classifiers generalized well to timepoints near the training window but poorly to more distal timepoints, with the generalization envelope widening as time progressed (4B). We again clustered the classifiers based on their temporal accuracy profile, then plotted mean profiles for each cluster (4 C). The result was an 'overlapping waves' pattern strikingly similar to the simulation: classifiers that performed well early in processing quickly declined in accuracy, replaced by a different well-performing set. Over time neighboring classifiers began to show similar temporal profiles, forming larger clusters that performed above chance for a broader temporal window (4D).

We next considered whether and how the neuro-semantic code changed over time. For each time window, we projected the classifier weights for all electrodes in all subjects to a cortical surface model, then animated the results (see *Video 1*). *Figure 4E* shows snapshots every 200 ms post stimulus onset. In mid-posterior regions, the code was spatially and temporally stable—weights on the lateral aspect were positive while those on the medial aspect were negative, consistent with feature-based views of representation. The anterior pattern differed, flipping from mainly positive at 200 ms to mainly negative by 800 ms and fluctuating across time and space throughout. In other words, the 'meaning' of a positive deflection in the voltage—whether it signaled animal or non-animal—stayed constant posteriorly but changed direction over time anteriorly, consistent with the deep, distributed, dynamic view (see A-3).

Finally, we assessed whether the four characteristics of the distributed and dynamic pattern noted earlier are statistically reliable in the ECoG data.

## Constant decodeability

The simulations suggest that semantic information should be decodeable from ATL neural activity from the onset of stimulus information and throughout stimulus processing. *Figure 4A* confirms this characteristic: from 200 ms post-onset, 95 % confidence intervals on mean decoding accuracy exceed chance for the duration of the stimulus-processing window.

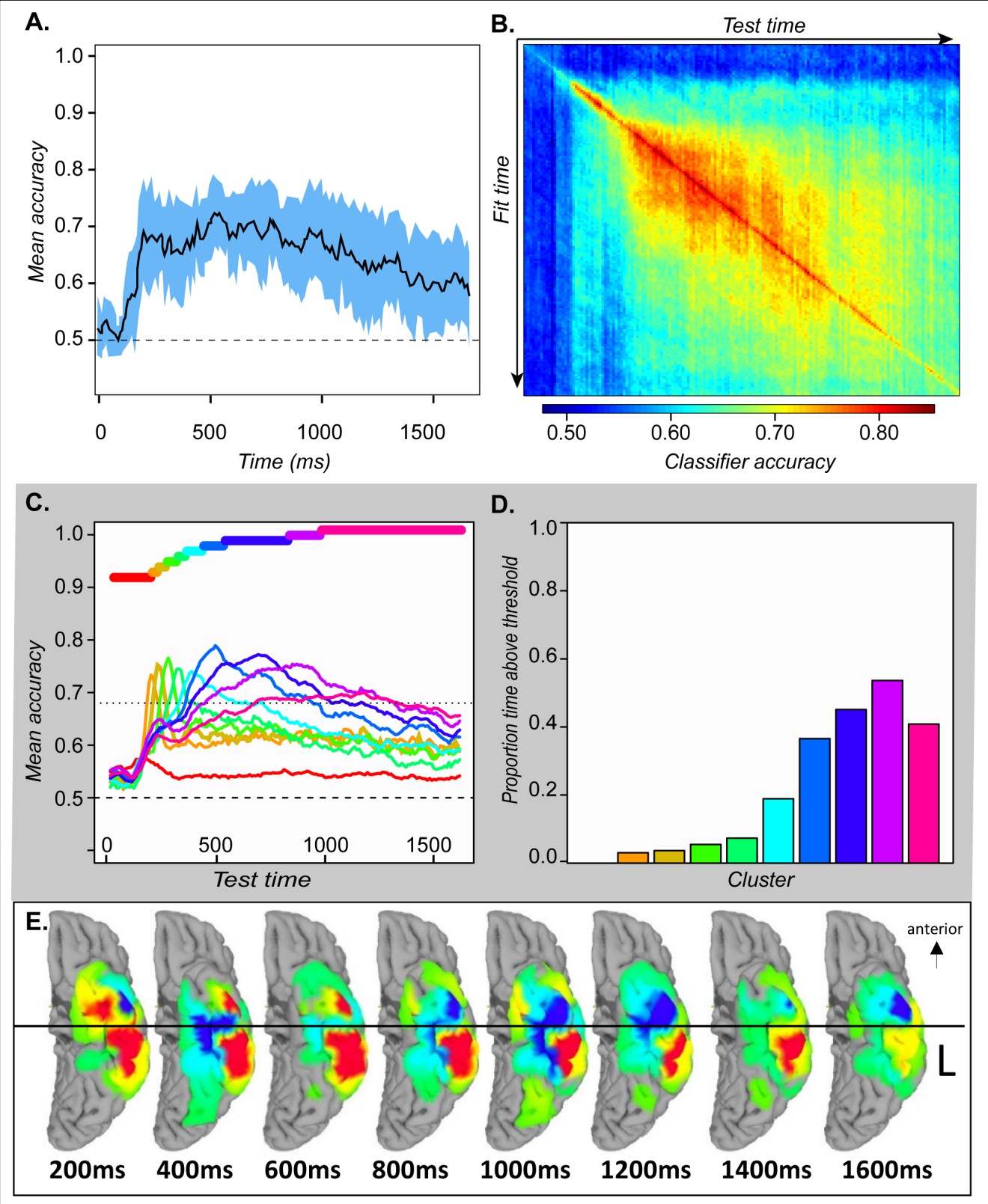

**Figure 4.** ECoG analyses. A. Mean and 95 % confidence interval of the hold-out accuracy for classifiers trained at each 50 ms time window of ECoG data. B. Mean accuracy across participants for each classifier (rows) tested at each timepoint (columns) in the ECoG data. C. Mean accuracy for each cluster of classifiers at every point in time. Colored bars show the timepoints grouped together in each cluster. D. Proportion of the full time-window for which mean classifier accuracy in each cluster was reliably above chance. E. Mean classifier coefficients across participants plotted on a cortical surface

*Figure 4 continued on next page*

*Figure 4 continued*

at regular intervals over the 1640 ms window. Warm vs cool colors indicate positive versus negative mean coefficients, respectively. In A and C, vertical line indicates mean onset of naming.

## Local temporal generalization

To assess whether the 'overlapping waves' pattern shown in *Figure 4C* is statistically reliable across subjects we conducted the following analysis. For each subject, we grouped the 160 classifiers into 10 clusters based on the preceding analysis, then computed the mean temporal decoding profile for each cluster separately in every subject. This provided a characterization of how well each cluster of classifiers performs over time in every individual subject. At each time-window, we identified the classifier-cluster that showed the highest hold-out accuracy across subjects, then used paired-samples t-tests to compute the probability that each remaining cluster was worse than the best-performing cluster. The probabilities were adjusted to control the false-discovery rate at $\alpha = 0.05$. *Figure 5A* shows the result. Yellow indicates the best-performing cluster and other statistically indistinguishable clusters. Darker colors indicate clusters that perform reliably worse than the best cluster at different levels of significance. Each cluster is reliably better than all other clusters at some point in time, excepting only the first (classifiers fit to early windows that never exceed chance) and the last (which performs compa-rably to the second-last cluster toward the end of the processing window). Thus, the 'overlapping waves' pattern shown qualitatively in *Figure 4C* reflects statistically reliable differences in classifier performance across subjects over time.

## Widening generalization window

.Simulations suggested that, as the system settles into an interpretation of the input, the neuro-semantic code grows more stable, so that classifiers generalize across a broader temporal window. *Figure 4D* shows a qualitatively similar pattern in ECoG; we conducted a further analysis to assess whether this widening pattern is statistically reliable. For classifiers fit at a given time window, we counted the number of test-windows where decoding accuracy reliably exceeded chance across subjects, using a statistical threshold of p < 0.01 uncorrected (ie about one expected false positive for the 160 comparisons). This provides an estimate of the 'width' of the temporal generalization envelope for classifiers fit at each time window. We then plotted generalization width against the time when the classifier was fit as shown by the gray dots in *Figure 5B*. The generalization window grows wider over the first ~500 ms of processing, where it hits a ceiling (ie classifiers generalize above chance for almost the whole window) then drops off for classifiers near the very end.

To assess the statistical reliability of this pattern, we fit a piecewise-linear regression to the data points shown in black in *Figure 5B*, which represent the set of 32 fully non-overlapping (and hence independent) time-windows beginning with the first. The piecewise linear approach first fits a linear model to the data, then successively adds breakpoints and uses the Bayes Information

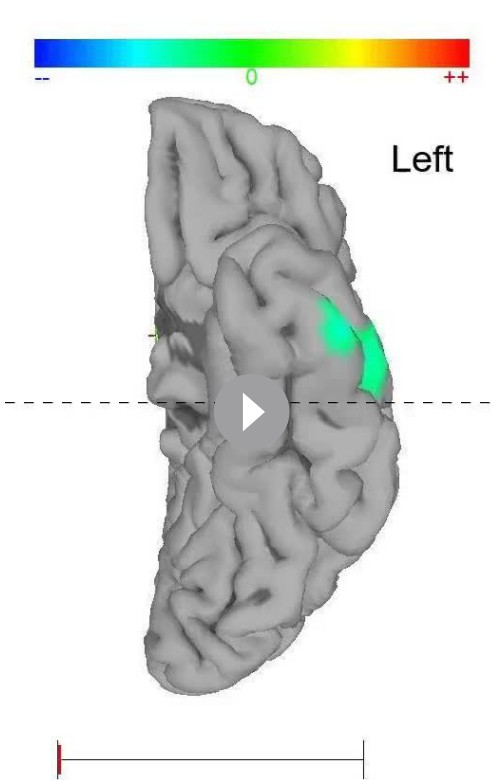

**Video 1.** Animation showing direction of classifier coefficients across all participants, projected and smoothed along the cortical surface, across successive time-windows over the course of stimulus processing. Colored regions indicate areas receiving non-zero coefficients, with cool colors indicating negative mean coefficients, green indicating means near zero, and warm colors indicating positive mean coefficients. Coefficients anterior to the dashed line fluctuate more relative to those posterior to the line, which are more consistent over time.

https://elifesciences.org/articles/66276/figures#video1

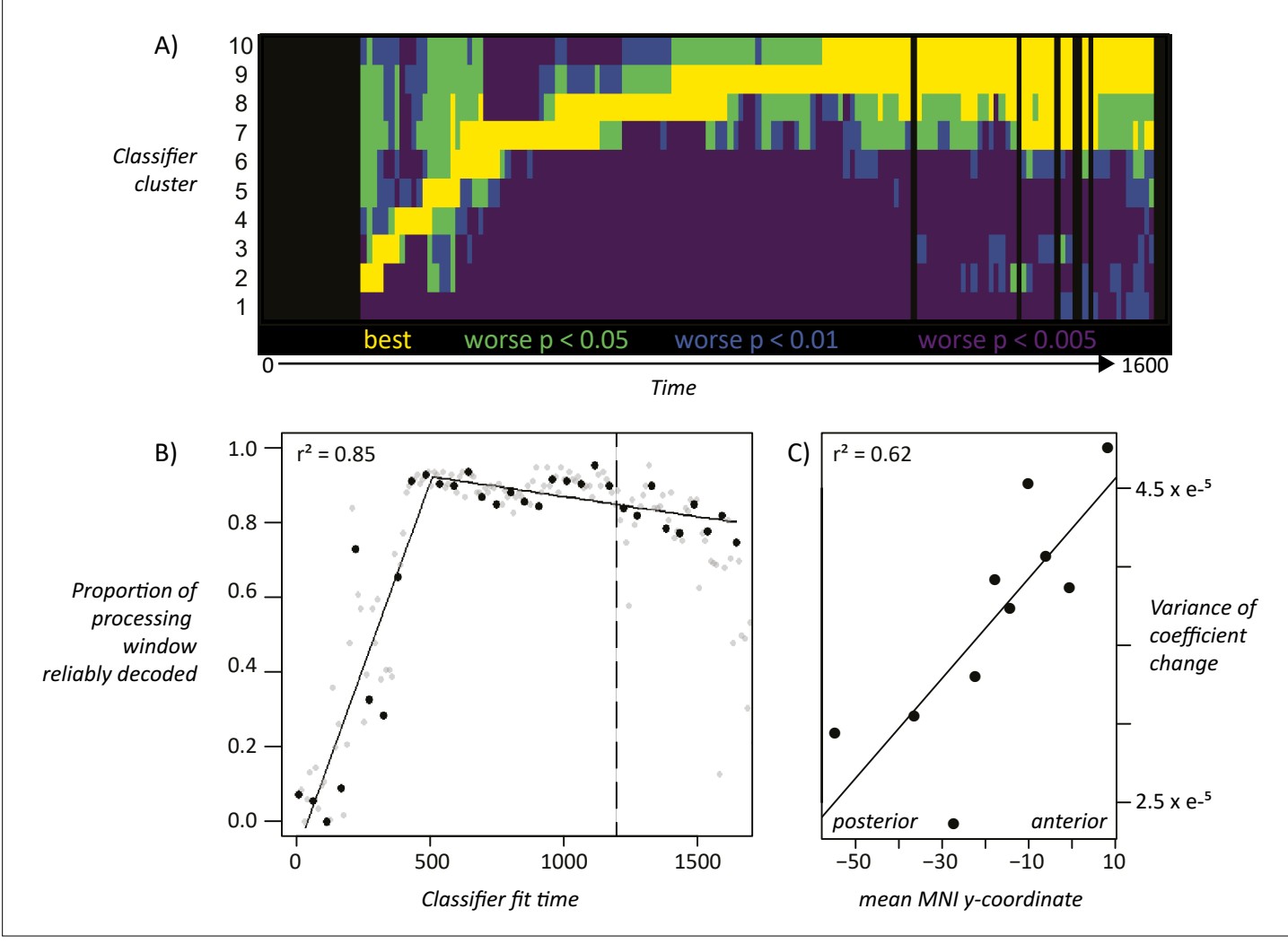

**Figure 5.** Statistical assessment of key patterns. A. *Statistical assessment of 'overlapping waves' pattern*. Each row corresponds to one cluster of decoding models as shown in *Figure 4C*. Black vertical lines indicate timepoints where decoding is not reliable across subjects. Yellow shows the best-performing model cluster and other clusters that statistically perform as well. Green, blue, and purple indicate clusters that perform reliably worse than the best-performing cluster at increasingly strict statistical thresholds controlling for a false-discovery rate of 0.05. B. *Broadening window of generalization*. For classifiers fit at each time window, breadth of classifier generalization (as proportion of full processing window) is plotted against the time at which the classifier was fit. The line shows a piecewise-linear model fit to 32 non-overlapping time windows (black dots). The most likely model had a single inflection point at 473 ms post stimulus-onset, with breadth of generalization increasing linearly over this span, then hitting ceiling through most of the remaining processing window. The dashed line shows mean response latency. C. *Fluctuating codes in more anterior regions*. Correlation between mean variance of coefficient change (see text) and anterior/posterior electrode location for electrodes grouped by decile along the anterior/posterior axis.

Criterion (BIC) to find the number of breakpoints that provides the best model fit given the number of free parameters. From this approach, the best model involved a single breakpoint at ~473 ms post stimulus onset. The black line shows the resulting bilinear model, which fits the data with $r^2 = 0.85$. The first model segment shows a reliable linear widening of the generalization window with fit time over ~500 ms ($r^2 = 0.70$, $p < 0.002$), providing strong evidence that the generalization window expands linearly with stimulus processing until it hits ceiling.

## Changing code direction

The surface maps shown in *Figure 4E* and accompanying animation (*Video 1*) suggest that classifier coefficients fluctuate more over time in anterior than posterior temporal cortex. Coefficient magnitude and direction indicate how a classifier 'interprets' deflections in the electrode voltage—that is,

whether a negative deflection signals 'living' or 'nonliving' in the context of other measured signals. The possibility that coefficients fluctuate more in anterior than posterior regions thus suggests that the anterior regions may show greater dynamic change in how they encode semantic information over time, as was also observed in the simulations.

To test whether this qualitative observation is statistically reliable, we measured the *variability of change* (VoC) in classifier coefficients at each electrode. For a given time window *t*, we took the difference between the coefficient for the classifier fit at *t* and the next fully non-overlapping time-window *t + 50* ms. This yielded, for each electrode, a series of 160 deltas indicating the magnitude and direction of coefficient change from one 50 ms time window to the next. We then computed, separately for each electrode, the *variance* of non-zero changes across this series. For electrodes whose coefficients change only a little, or change consistently in the same direction (e.g. linearly ramping up over time), the VoC metric is small; for coefficients that change substantially in magnitude and/or direction, it is large. To determine whether VoC differs reliably along the anterior/posterior axis of the temporal lobe, we divided all electrodes across subjects into deciles based on the anterior/posterior dimension of their MNI coordinate. We then computed the mean VoC for each decile, and regressed it on the mean anterior/posterior MNI coordinate for the decile. The result is shown in *Figure 5C*: anterior/posterior location predicts VoC with $r^2 = 0.73$, $P < 0.002$.

To assess whether this pattern is reliable across participants, we fit a separate linear model for every participant predicting VoC from electrode anterior/posterior MNI coordinate, and recorded the estimated slope of the relationship for each. Across participants this slope was reliably positive ($t(7) = 2.58$ vs null hypothesis of zero slope, $p < 0.02$ one-tailed), indicating greater fluctuation of classifier coefficients in more anterior regions across participants.

## Discussion

We have combined computational modeling, multivariate pattern classification, and human ECoG to better understand how ventral temporal cortex encodes animacy information about visually-presented stimuli. The results suggest that, at the scale of intracranial EEG, neural signals throughout ventral temporal lobe jointly express such information from about 200 ms post stimulus-onset, via a distributed code that is stable and feature-like in middle/posterior regions but dynamic and nonlinearly-changing in anterior regions. In simulation we showed that similar phenomena arise in a deep, interactive neuro-semantic model, producing a characteristic decoding signature: classifiers perform well in the time-window when they were trained, but generalize over a narrow time envelope that widens as the system settles. This pattern was only observed in a model combining distributed representation, interactive processing, and a deep architecture (see Appendix). Remarkably similar phenomena were observed in ECoG data collected from the surface of ventral temporal lobe while participants named line-drawings, supporting the proposal that vATL encodes distributed semantic representations that change dynamically and nonlinearly with stimulus processing by virtue of this region's role as a deep hub within the cortical semantic network.

This proposal resolves a long-standing puzzle. Convergent methods have established the centrality of vATL for semantic memory, including studies of semantic impairment (*Julie et al., 1989*; *Hodges et al., 1995*; *Lambon Ralph et al., 2007*), lesion-symptom mapping (*Acosta-Cabronero et al., 2011*), functional (*Visser et al., 2010*; *Rogers et al., 2006*) and structural (*Binney et al., 2012*; *Binney et al., 2010*) brain imaging, and transcranial magnetic stimulation (*Pobric et al., 2010*; *Pobric et al., 2007*). Yet multivariate approaches to discovering neuro-semantic representations rarely identify the vATL, instead revealing semantic structure more posteriorly (*Bruffaerts et al., 2013*; *Devereux et al., 2013*; *Sha et al., 2015*). One prominent study suggested that semantic representations may tile the entire cortex *except* for the vATL (*Huth et al., 2016*). Setting aside significant technical challenges of successful neuroimaging of this region (*Binney et al., 2010*), almost all such studies have employed non-invasive imaging techniques that sacrifice either temporal or spatial resolution—a compromise that will destroy signal in vATL if semantic representations there are distributed and dynamic at the scale we have measured, but will preserve signal in posterior regions where the code is more stable. Thus the widespread null result may arise precisely because semantic representations in vATL are distributed and dynamic.

Consistent with this view, the current results in middle/posterior regions accord well with evidence from classical functional brain imaging of visual semantics. Several fMRI studies have reported

category effects in the mid-posterior fusiform, with animate items eliciting greater activation on the lateral bank and inanimate items eliciting greater activation on the medial bank (*Martin and Chao, 2001*; *Anzellotti et al., 2011*; *Mahon et al., 2009*). In our analysis, these are regions where a stable, feature-like code arises, with animate items signaled by positive voltages on the lateral aspect and negative voltages on the medial aspect. Since the location and direction of the code are more stable in these regions within and across subjects, the signal remains detectable even with the spatial and temporal averaging that occurs in univariate fMRI.

These conclusions rest partly on the analysis of classifier weights. When multiple channels are affected by a common source of noise, multivariate decoders can place weights on uninformative channels that 'subtract out' the noise corrupting the signal on informative channels (*Haufe et al., 2014*). Might the fluctuating weights observed in vATL arise, not because the region encodes semantic structure, but because it serves to 'cancel out' effects of correlated noise on the signal-carrying channels more posteriorly? Four lines of evidence suggest not. First, we fit classifiers to only the anterior electrodes in each participant and observed reliable decoding—indeed, performance was equally reliable for classifiers trained on anterior-only and posterior-only datasets (A-4). Second, an earlier study applied searchlight representational similarity analysis (RSA) to the same data (*Chen et al., 2016*), and found that vATL was the only local region that reliably encodes semantic similarity structure. Neither result could obtain if vATL simply subtracted out correlated noise from more posterior areas. Third, the observed pattern of a stable code posteriorly and fluctuating code anteriorly was predicted by the current simulations, using a model validated against neuropsychological, anatomical, and brain-imaging results in prior work (*Patterson et al., 2007*). Fourth, the critical importance of ATL for semantic representation has been established by the broad range of converging evidence cited previously.

Prior studies applying temporal generalization to MEG data in visual semantic tasks uniformly report a very narrow and unchanging band of temporal generalization (*Carlson et al., 2013*; *Cichy et al., 2014*), a pattern consistent with the proposal that the neuro-semantic code changes rapidly over the course of stimulus processing. Our results differ from the MEG pattern, and indeed from most other work applying the temporal generalization approach (*King and Dehaene, 2014*), in showing a gradual widening of the temporal generalization window. This phenomenon does not arise from the autocorrelational structure of the data itself—the window of time over which an electrode reliably predicts its own future state does not grow wider with stimulus processing (A-5). Instead the widening must reflect an increasingly stable representational code. The simulation explains why the pattern arises in anterior temporal cortex: hub representations in vATL change rapidly early on due to interactions with modality-specific representations throughout cortex, but these changes slow as the full activation pattern emerges across network components.

## Limitations and future directions
### Is the animacy code truly semantic?
We have focused on decoding a broad, binary semantic distinction that is a common focus of much work in this area—specifically, whether an image depicts an animate or an inanimate item. Animacy is a useful starting point because it is not transparently captured by low-level perceptual structure; in our stimuli, for instance, low-level visual similarity as expressed by Chamfer matching does not reliably distinguish the animate and inanimate items (see Materials and Methods). Nevertheless it remains possible that decoders in the current work exploit some other property that happens to be confounded with animacy. Whether this is the case or not, the preceding arguments suggest that the relevant information is expressed in a distributed, dynamically-changing neural code.

Further evidence for semantic structure could involve decoding the graded within- and between-domain conceptual similarities existing amongst the stimuli. The question of how to fit such a decoder is, however, somewhat complex, with no standard solution. Common unsupervised methods like representational similarity analysis—where one computes the correlation between neural and target dissimilarity matrices—don't fit the bill, because such correlations can yield a positive result even if the signal or target truly encodes just a binary label (as shown, e.g., in 45), and can yield a negative result if a signal is present but buried in amongst many irrelevant features (*Cox and Rogers, 2021*). "Generative" approaches (e.g. *Pereira et al., 2018* )—where one predicts the neural response at each electrode from a feature-based description of the stimulus—are problematic because they assume that

each element of the neural code is independent (ie, the "meaning" of a voltage deflection at a given electrode is the same regardless of the states of other electrodes). As outlined in the introduction, our aim is to test the possibility that the neural code is distributed in a manner consistent with neural network models, where the "meaning" of a given unit varies depending on the states of other units.

Ideally one wants a multivariate decoding model that can be fit to all data within a given time window without feature preselection (as in the current paper), but which predicts the embedding of stimuli within a multidimensional semantic space instead of a simple binary classification. *Oswal et al., 2016* recently developed theory and analysis for such an approach tailored to the decoding of neural data, and future work should consider whether more continuous/fine-grained aspects of semantic similarity can be decoded from vATL using such techniques.

### Decoding voltage versus time-frequency spectrograms

We have taken the voltage measured at an electrode as a neural analog of unit activation in an artificial neural network model. It remains unclear, however, how the processing units in a neural network are best related to the signals measured by ECoG. Voltages measured at a surface electrode can be influenced by firing of nearby neurons but also by the activity of incoming synapses from neural populations at varying distances from the recording sites. Thus there is no guarantee that signals measured in vATL only reflect neural activation arising in vATL. Many ECoG studies decompose voltage time-series into time-frequency spectrograms, which indicate the power of oscillations arising in the signal at different temporal frequencies. Power in the gamma band is often thought to reflect spiking activity of local neurons (*Merker, 2013*), and thus might provide a better indication of the activity of neurons within vATL proper.

We have not undertaken such an analysis for two reasons. First, our goal was to assess whether the neural code for animacy has the distributed and dynamic properties that arise within a well-studied neural network model of semantic cognition. Such models do not exhibit oscillating behavior, making the model analog to frequency bands unclear. Second, Prior ECoG work has shown that, while object- and person-naming does elicit increased gamma for more perceptual areas (like posterior fusiform), in anterior temporal regions it significantly alters beta-band power (*Abel et al., 2016*). Similarly, (*Tsuchiya et al., 2008*) applied multivariate decoding to ECOG signals to discriminate happy and fearful faces, finding reliable information in ventral temporal cortex in frequencies < 30 Hz (as well as in gamma). Such patterns may reflect ATL's connections to multiple distal cortical areas, since beta-band oscillations are thought to aid in coordination of long-distance neural interactions (*Kopell et al., 2000*). Thus we have left the decoding of time-frequency information in these signals, and their connection to hypothesized information processing mechanisms in neural network models, to future work.

### Limitation to ventral ATL

We have focused on vATL because of its widely-recognized role in semantic cognition (*McClelland and Rogers, 2003*; *Patterson et al., 2007*; *Lambon Ralph et al., 2017*). A drawback of the current approach is its limited field of view: we cannot draw inferences about other parts of cortex involved in semantic processing of visually-presented images, or whether the code arising in such areas changes dynamically. The current data do not, for instance, address important questions about the role of parietal, frontal and lateral/superior-temporal systems in semantic processing. The temporal generalization methods we have adopted may, however, contribute to these questions if applied to datasets collected from electrodes situated in these regions in future work.

## Conclusion

Why should a dynamic distributed code arise specifically within the vATL? The area is situated at the top of the ventral visual stream, but also connects directly to core language areas (*Nobre et al., 1994*; *Mesulam, 1998*) and, via middle temporal gyrus, to parietal areas involved in object-directed action (*Chen et al., 2017*). It receives direct input from smell and taste cortices (*Gloor, 1997*), and is intimately connected with limbic structures involved in emotion, memory, and social cognition (*Gloor, 1997*). Thus vATL anatomically forms the hub of a cross-modal network ideal for encoding associations among visual, linguistic, action, sensory, and social/motivational representations. Hub neurons interact with a wide variety of subsystems, each encoding a different kind of structure and content,

potentially pushing the hub representations in different directions over time as activity propagates in the network. Other network components lying closer to the sensory or motor periphery connect mainly within individual modality-specific systems (*Binney et al., 2012*), so may be less impacted by such cross-modal interactions, as observed in the model. For this reason, the feature-based approach that has proven indispensable for characterizing neural representations in modality-specific cortices may be less suited to understanding the distributed and dynamic representations that arise in deeper and more broadly-connected (tertiary association) cortical regions. These include regions critical for human semantic knowledge, and potentially other higher-level cognitive functions.

## Materials and methods

### Simulation study

The model implements the 'distributed-plus-hub' theory of semantic representation developed in prior work (*Patterson et al., 2007*; *Lambon Ralph et al., 2017*; *Rogers et al., 2004*; *Chen et al., 2017*). It is a deep, fully continuous and recurrent neural network that learns associations among visual representations of objects, their names, and verbal descriptors, via a central cross-modal hub, with units and connectivity shown in *Figure 2A* of the main paper. Simulations were conducted using the open-source Light Efficient Network Simulator (*Rohde, 1999*) updated for contemporary libraries (https://github.com/crcox/lens). Code for replicating the simulations and the data reported in this paper are available at https://github.com/ttrogers, (copy archived at swh:1:rev:8c-c85427c81922d107e33ef58f038fd228f042fe; RRID: SCR_021099).

All units employed a continuous-time sigmoidal activation function with a time-constant of 0.25. Visual and Verbal units were given a fixed, untrainable bias of –3 that produced a low activation state without positive input. Hidden units had trainable biases. To simulate perception of an image, units encoding the item's visual representation were given direct positive stimulation of +6 so that, combined with the fixed bias, they received a net input of +3 (in addition to any inputs from other units in the model) which persisted through stimulus processing. The resulting changes in unit activations propagated through visual hidden, hub, and verbal hidden units to eventually alter activation states in the verbal units. Because the model was reciprocally connected, such downstream changes fed back to influence upstream states at each moment of simulated time, as the whole system settled to a stable state. To simulate verbal comprehension, the same process unfolded, but with positive input externally provided to verbal units. Units updated their activation states asynchronously in permuted order on each tick of time and were permitted to settle for five time intervals (a total of 20 updates) during training and eight time intervals (32 updates) during testing.

### Model environment

The model environment contained visual and verbal patterns for each of 90 simulated objects, conceived as belonging to three distinct domains (e.g. animals, objects, and plants). Each domain contained 10 items from each of three sub-categories—thus there were 30 'animals', 30 'objects' and 30 'plants'. Visual patterns were constructed to represent each item by randomly flipping the bits of a binary category prototype vector in which items from the same domain shared a few properties and items from the same category shared many. The verbal patterns were constructed by giving each item a superordinate label true of all items within a given domain (animal, object, plant), a basic-level label true of all items within a category (e.g. 'bird', 'fish', 'flower', etc), and a subordinate label unique to the item (e.g. 'robin', 'salmon', 'daisy', etc). These procedures, adopted from prior work (*Rogers et al., 2004*), generated model input/target vectors that approximate the hierarchical relations among natural concepts in a simplified manner that permits clear understanding and control of the relevant structure.

### Training

For each input, target patterns that fully specified the item's visual and verbal characteristics were applied throughout the duration of stimulus processing. The model was trained with backpropagation to minimize squared error loss. Half of the training patterns involved generating verbal outputs from visual inputs, while the other half involved generating visual outputs from verbal inputs. The model was initialized with small random weights sampled from a uniform distribution ranging from –1–1, then

**Table 1.** Patient characteristics.

CPS: complex partial seizure; GTCS: generalized tonic clonic seizure; ECoG: electrocorticogram; ERS: epigastric rising sensation; a/pMTG: anterior/posterior part of the middle temporal gyrus; a/pMTG: anterior/posterior part of the middle temporal gyrus; FCD: focal cortical dysplasia; * dual pathology ** diagnosed by clinical findings.

| | Patient 1 | Patient 2 | Patient 3 | Patient 4 |
|---|---|---|---|---|
| Age, gender, handedness | 22 M R | 29 M R&L | 17 F R | 38 F R |
| WAIS-R (VIQ,PIQ,TIQ) | 70, 78, 69 | 72, 78, 72 | 67, 76, 69 | 84,97,89 |
| WMS-R (Verb, Vis, Gen, Attn, Del recall) | 99, 64, 87, 91, 82 | 99, 92, 97, 87, 83 | 51, < 50, < 50, 81, 56 | 75,111,83,62,53 |
| WAB | 95.6 | 96 | 97.2 | 98.5 |
| WADA test (Language) | Left | Bilateral | Left | Left |
| Age of seizure onset | 16 | 10 | 12 | 29 |
| Seizure type | non-specific aura → CPS, GTCS | aura (metamorphosia, ERS) → CPS | discomfort in throat → CPS | ERS →CPS |
| Ictal ECoG onset | aMTG | PHG | PHG | PHG |
| MRI | L basal frontal cortical dysplasia L anterior temporal arachnoid cyst | L posterior temporal cortical atrophy | L temporal tip arachnoid cyst | L hippocampal atrophy/ sclerosis |
| Pathology | FCD type IA | FCD type IA Hippocampal sclerosis* | FCD type IB | Hippocampal sclerosis** |

| | Patient 5 | Patient 7 | Patient 9 | Patient 10 |
|---|---|---|---|---|
| Age, gender, handedness | 55 M R | 41 F R | 51 M R | 38 F R |
| WAIS-R (VIQ,PIQ,TIQ) | 105,99,103 | 72, 83, 75 | 73, 97, 83 | 109, 115,112 |
| WMS-R (Verb, Vis, Gen, Attn, Del recall) | 71,117,84,109,72 | 83,111,89,94,82 | 80,101,85,1919,91 | 71,79,70,90,58 |
| WAB | 98 | 97.3 | 89.6 | 96.9 |
| WADA test (Language) | Left | Right | Left | Left |
| Age of seizure onset | 55 | 19 | 43 | 28 |
| Seizure type | CPS (once) | aura (nausea,feeling pale) → CPS | CPS | non-specific aura → CPS |
| Ictal ECoG onset | none | PHG | mITG | SMG |
| MRI | Low-grade glioma L medial temporal lobe | L hippocampal atrophy/screrosis L parieto-occipital perinatal infarction | Left temporal cavernoma | L parietal opercurum tumor |
| Pathology | Diffuse astrocytoma | FCD IA Hippocampal sclerosis* | arteriovenous malformation | Oligoastrocytoma |

trained for 30,000 epochs in full batch mode with a learning rate of 0.002 and without weight decay. For each pattern, the settling process was halted after 20 activation updates, or when all Visual and Verbal units were within 0.2 of their target values, whichever came first. For all reported simulations, the model was trained five times with different random weight initializations. After training, all models generated correct output activations (i.e. on the correct side of the unit midpoint) for more than 99 % of output units across all training runs. Each model was analyzed independently, and the final results were then averaged across the five runs.

## Testing

The picture-naming study was simulated by presenting visual input for each item, recording the resulting activations across the 25 hub units at each update as the model settled over 32 updates, and distorting these with uniform noise sampled from –0.005 to 0.005 to simulate measurement error. As the model settles over time it gradually activates the word unit corresponding to the item name, and in this sense simulates picture naming. Just as in the ECoG data, we recorded unit activations across a fixed period of time, regardless of when the correct name unit became active.

Note that, whereas the ECoG study employed items drawn from two general semantic domains (living and nonliving), the model was trained on three domains. This provided a simple model analog to the true state of affairs in which people know about more semantic kinds than just those appearing in the ECoG stimulus set. To simulate the study, the model was presented with 60 items selected equally from two of the three semantic domains—so as in the study, half the stimuli belonged to one domain and half to another. To ensure results did not reflect idiosyncrasies of one domain, we simulated the task with each pair of domains and averaged results across these.

## Analysis

All analyses were conducted using R version 3.6. To visualize the trajectory of hub representations through unit activation space as a stimulus is processed, we computed a simultaneous three-component multidimensional scaling of the unit activation patterns for all 90 items at all 33 timepoints using the native R function cmdscale. The resulting coordinates for a given item at each point in time over the course of settling were plotted as lines in a 3D space using the scatterplot3d package in R. *Figure 2B* shows the result for one network training run. *Figure 2C* shows the same trajectories in the raw data (i.e. actual unit activation states rather than latent dimensions in a MDS) for randomly sampled pairs of hub units.

To simulate decoding of ECoG data, we evaluated logistic classifiers in their ability to discriminate superordinate semantic category from patterns of activity arising in the hub at each timepoint. As explained in the main text, we assume that ECoG measures only a small proportion of all the neural populations that encode semantic information. We therefore sub-sampled the hub-unit activation patterns by selecting three units at random from the 25 hub units and using their activations to provide input to the decoder. Classifiers were fitted using the glm function and the binomial family in R. A separate decoder was fitted at each time-point, and unit activations were mean-centered independently at each time point prior to fitting. We assessed decoder accuracy at the time-point where it was fitted using leave-one-out cross-validation, and also assessed each decoder at every other time point by using it to predict the most likely stimulus category given the activation pattern at that time point and comparing the prediction to the true label. This process was repeated 10 times for each model with a different random sample of three hub units on each iteration. The reported results then show mean decoding accuracy averaged over the five independent network training runs, for decoders trained and tested at all 33 time points. The above procedure yielded the decoding accuracy matrix shown as a heat plot in *Figure 3B*.

Each row of this matrix shows the mean accuracy of decoders trained at a given timepoint, when those decoders are used to predict item domain at each possible timepoint. The diagonal shows hold-out accuracy for decoders at the same time point when they are trained, but off-diagonal elements show how the decoders fare for earlier (below diagonal) or later (above) timepoints. Decoders that perform similarly over time likely exploit similar information in the underlying representation, and so can be grouped together and their accuracy profiles averaged to provide a clearer sense of when the decoders are performing well. To this end, we clustered the rows of the decoding accuracy matrix by computing the pairwise cosine distance between these and subjecting the resulting similarities

to a hierarchical clustering algorithm using the native hclust function in R with complete agglomeration. We cut the resulting tree to create 10 clusters, then averaged the corresponding rows of the decoding accuracy matrix to create a temporal decoding profile for each cluster (lines in *Figure 3C*). We selected 10 clusters because this was the highest number in which each cluster beyond the first yielded a mean classification accuracy higher than the others at some point in time. Similar results were obtained for all cluster-sizes examined, however.

Finally, to understand the time-window over which each cluster of decoders performs reliably better than chance, we computed a significance threshold using a one-tailed binomial probability distribution with Bonferroni correction. Each decoder discriminates two categories from 60 items, with probability 0.5 of membership in either category. We therefore adopted a significance threshold of 44 correct items out of 60, corresponding to a binomial probability of $p < 0.03$ with Bonferroni correction for 330 tests (10 clusters at each of 33 time points). The barplot in *Figure 3D* shows the proportion of the full time window during which each decoding cluster showed accuracy above this threshold.

## ECoG study

### Participants

Eight patients with intractable partial epilepsy (seven) or brain tumor (one) originating in the left hemisphere participated in this study. These include all left-hemisphere cases described in a previous study (*Chen et al., 2016*), and we will use the same case numbers reported in that work (specifically cases 1–5, 7, and 9–10). Background clinical information about each patient is summarized in *Table 1*. Subdural electrode implantation was performed in the left hemisphere for presurgical evaluation (mean 83 electrodes, range 56–107 electrodes/patient). A total of 16–24 electrodes (mean 20 electrodes) covered the ventral ATL in each patient. The subdural electrodes were constructed of platinum with an inter-electrode distance of 1 cm and recording diameter of 2.3 mm (ADTECH, WI). ECoG recording with subdural electrodes revealed that all epilepsy patients had seizure onset zone outside the anterior fusiform region, except one patient for whom it was not possible to localize the core seizure onset region. The study was approved by the ethics committee of the Kyoto University Graduate School of Medicine (No. C533). Participants all gave written information consent to participate in the study.

### Stimuli and procedure

One hundred line drawings were obtained from previous norming studies (*Barry et al., 2018*; *Snodgrass and Vanderwart, 1980*), including 50 items in each of the two semantic categories that were the target of our decoding efforts (animal vs non-animal). A complete list of all items can be found in *Chen et al., 2016*. Living and nonliving stimuli were matched on age of acquisition, visual complexity, familiarity and word frequency, and had high name agreement. Independent-sample t-tests did not reveal any significant differences between living and nonliving items for any of these variables.

To assess whether animal / non-animal stimuli differed systematically in their non-semantic visual structure, we computed pairwise visual similarities using Chamfer matching, an unsupervised technique from machine vision suited to aligning and then characterizing the similarity between pairs of bitmapped images (*Barrow et al., 1977*). We computed these similarities for all pairs of images. In the resulting matrix, each image corresponds to a row-vector of similarities. We then assessed whether it is possible to reliably decode living/nonliving from these vectors, using L1-regularized logistic regression and 10-fold cross-validation, exactly as was done for the ECoG data. Mean accuracy across folds was 0.51 and did not differ reliably from chance, indicating that the animal/non-animal distinction is not readily discernable solely from the non-semantic visual structure of the images.

Participants were presented with stimuli on a PC screen and asked to name each item as quickly and accurately as possible. All stimuli were presented once in a random order in each session and repeated over four sessions in the entire experiment. The responses of participants were monitored by video recording. Each trial was time-locked to the picture onset using in-house MATLAB scripts (version 2010 a, Mathworks, Natick, MA). Stimuli were presented for 5 seconds each and each session lasted 8 minutes 20 seconds. Participants' mean naming time was 1190 ms. Responses and eye fixation were monitored by video recording.

## Data preprocessing

Data preprocessing was performed in MATLAB. Raw data were recorded at sampling rate of 1000 Hz for six patients and at 2000 Hz for two patients. The higher sampling rates for the two patients were down-sampled to 1000 Hz by averaging measurements from each successive pair of time-points. The raw data from the target subdural electrodes for the subsequent analysis were measured in reference to the electrode beneath the galea aponeurotica in four patients (Patients 4,5,7 and 10) and to the scalp electrode on the mastoid process contralateral to the side of electrode implantation in four patients (Patients 1–3 and 9). Multivariate pattern-classification analyses were also conducted without such referencing and yielded near-identical results. Baseline correction was performed by subtracting the mean pre-stimulus baseline amplitude (200 ms before picture onset) from all data points in the epochs. Trials with greater than±500 µV maximum amplitude were rejected as artifacts. Visual inspection of all raw trials was conducted to reject any further trials contaminated by artifacts, including canonical interictal epileptiform discharges. The mean waveform for each stimulus was computed across repetitions. Data included, for each stimulus at each electrode, all measurements beginning at stimulus onset and continuing for 1640 ms. While this window includes the onset of articulation toward the end, the critical results cannot reflect such motor activity since all key phenomena are observed prior to mean time to initiate the utterance (1190 ms).

## Multivariate classification analysis

The pre-processed data yielded, for each electrode in each patient, a times-series of voltages sampled at 1000 Hz over 1640 ms for each of 100 stimuli. For each patient, we trained classifiers to discriminate animal from non-animal images given the voltages evoked by each stimulus across all ventral-temporal electrodes in a 50 ms time-window. In a patient with 20 electrodes, one 50 ms window contains 1,000 measurements (50 voltages for each electrode x 20 electrodes). For each time window in every patient, these measurements were concatenated into feature vectors for each of the 100 stimuli, with the time windows advancing along the time-series in 10 ms steps. Thus, the first window included 1–50 ms post stimulus onset, the next included 11–60 ms, and so on. This procedure yielded feature vectors for all 100 items in 160 time-windows for every subject.

The classifiers were logistic regression models fitted with L1 regularization (*Tibshirani, 1996*) using the glmnet function in Matlab. L1-regularization applies an optimization penalty that scales with the sum of the absolute value of the classifier coefficients and thereby encourages solutions in which many features receive coefficients of 0. This approach is useful for a sliding-window analysis because features receiving a 0 coefficient in the classifier have no impact on its performance when it is assessed at other time points. So long as the information exploited by a classifier at time *t* is present at a different time *t±n*, the classifier will continue to perform well, even if other features are in very different states. Thus, classifiers trained with L1 regularization have the potential to show dynamic changes in the underlying code.

Classifier accuracy for a given time-window and subject was assessed using nested 10-fold cross-validation. In each outer fold, 10 % of the data were held out, and the remaining 90 % of the data were used with standard 9-fold cross-validation to search a range of values for the regularization parameter. When the best weight was selected, a model was fitted to all observations in the 90 % of the training data and evaluated against the remaining 10 % in the outer-loop hold-out set. This process was repeated 10 times with different final hold-outs, and classifier accuracy for each patient was taken as the mean hold-out accuracy across these folds. The means across patients are the data shown in *Figure 4A* and the diagonal of 4B in the main paper. A final classifier for the window was then fitted using all of the data and the best regularization parameter. This classifier was used to decode all other time-windows, yielding the off-diagonal accuracy values shown in *Figure 4B*.

The above procedures produced a pattern classifier for each of 160 50 ms time-windows in every subject, with every classifier then tested at every time-window within each subject. Thus, the classifier accuracy data were encoded in a 160 × 160-element decoding matrix in each subject. The matrices were averaged to create a single 160 × 160-element matrix indicating the mean decoding accuracy for each classifier at each point in time across subjects. This is the matrix shown in *Figure 4B*.

To better visualize how the code exploited by each classifier changes over time, we clustered the rows using the same agglomerative hierarchical approach described for the simulations. We considered solutions ranging from 4 to 15 clusters and plotted the mean decoding accuracy over time across

the classifiers within each cluster. All cluster sizes produced the overlapping-waves pattern. In the main paper, we show the 10-cluster solution as it is the largest number in which each cluster after the first has a mean accuracy profile that is both statistically reliable and higher than every other cluster at some point in time.

To assess the breadth of time over which a cluster showed reliable above-chance classification accuracy, we again set Bonferroni-corrected significance thresholds using the binomial distribution. Stimuli included 100 items, with a .5 probability of each item depicting an animal. In the 1640ms measurement period there are 32 independent (i.e. non-overlapping) 50 ms time windows, and we assessed the mean classifier performance for each of 10 clusters at every window. We therefore corrected for 320 multiple comparisons using a significance threshold of 68 correct (p < 0.0001 per comparison, p < 0.03 with correction; this is the lowest number correct that yield a corrected p value less than 0.05).

## Surface plot visualization

Magnetization-prepared rapid gradient-echo (MPRAGE) volumetric scan was performed before and after implantation of subdural electrodes as a part of presurgical evaluations. In the volumetric scan taken after implantation, the location of each electrode was identified on the 2D slices using its signal void due to the property of platinum alloy. Electrodes were non-linearly co-registered to the patient MRI (MPRAGE) taken before implantation, and then to MNI standard space (ICBM-152) using FNIRT (https://www.fmrib.ox.ac.uk/fsl/fnirt/). The native coordinates of all the electrodes for all patients were morphed into MNI space and resampled into 2 mm isotropic voxels55.

## Projecting classifier coefficients to the surface

As described above, a separate logistic classifier was fitted to each 50 ms window in each subject. The classifier was specified as a set of regression coefficients, with one coefficient for each timepoint at each electrode in the patient, and many coefficients set to 0 due to L1-regularization. The sign of the classifier coefficient indicates the 'meaning' of a oltage deflection in a particular direction: a positive coefficient indicates that animals are 'signaled' by a positive deflection in the voltage, while negative coefficients indicate that animals are signaled by a negative deflection. The magnitude of the coefficient indicates the 'importance' of the measurement, in the context of all other voltages appearing in the classifier. The distribution of coefficient directions and magnitudes across the cortex and over time thus provides an indication of how the underlying neuro-semantic code changes over time.

For a single time window we computed, separately for each electrode in each participant, the magnitudes (sum of absolute values) of the classifier weights across the 50 time points in the window. The resulting data were exported from Matlab to NIFTI volumes using the NIFTI toolbox (https://www.mathworks.com/matlabcentral/fileexchange/8797-tools-for-nifti-and-analyze-image) and projected from all electrodes and subjects onto the common cortical surface map using AFNI's 3dVol2Surf relative to the smooth white matter and pial surfaces of the ICBM 152 surface reconstructions shared by the AFNI team and the NIH (https://afni.nimh.nih.gov/ pub/dist/tgz/suma_MNI152_2009.tgz). The space between corresponding nodes on the two surfaces were spanned by a line segment subdivided at 10 equally spaced points. The value displayed on the surface is the average of the values where these 10 points intersect with the functional volume along that line segment. Once mapped to the surface, the results were spatially smoothed along the surface with an 8 mm full-width half-max Gaussian kernel using the SurfSmooth function in SUMA. We inclusively masked any surface point with a non-zero value in this surface projection. A separate mask was generated for each time window.

To visualize how the representational code changes over time within the surface mask, we next carried out a similar procedure on the classifier coefficients themselves, without taking the absolute values. At each electrode in every subject we summed the classifier coefficients over the 50 ms time window, yielding a single positive or negative real-valued number at each electrode for each time window. These values were again projected onto a common brain surface and spatially smoothed with an 8 mm FWHM Gaussian blur along the surface. In the resulting maps, any colored point indicates a cortical region that received a non-zero value in the weight magnitude mask, while the hue indicates the direction of the classifier coefficient in the area—that is, whether a positive deflection of the voltage for nearby electrodes indicated that the stimulus was an animal (warm colors), a nonanimal (cool colors), or showed no systematic direction (green). A separate map of this kind was generated for each of 160 time windows. We animated the results to visualize how they change over

time using the open-source ffmpeg software (https://ffmpeg.org/) with linear interpolation between successive frames. The animation is shown in *Video 1*; snapshots of this visualization are shown in *Figure 4E*.

## Acknowledgements

This work was partially supported by Medical Research Council Programme Grant MR/R023883/1, intra-mural funding MC_UU_00005/18 and by European Research Council grant GAP: 502670428 - BRAIN2MIND_NEUROCOMP

## Additional information

### Competing interests

Akio Ikeda: The Department of Epilepsy, Movement Disorders, and Physiology is an Industry-Academia Collaboration Course, supported by Eisai Co, Ltd, Nihon Kohden, Corporation, Otsuka Pharmaceutical Co, and UCB Japan Co, Ltd. The other authors declare that no competing interests exist.

### Funding

| Funder | Grant reference number | Author |
| --- | --- | --- |
| Medical Research Council | MR/J004146/1 | Matthew A Lambon Ralph |
| European Research Council | GAP: 502670428 - BRAIN2MIND_ NEUROCOMP | Matthew A Lambon Ralph |

The funders had no role in study design, data collection and interpretation, or the decision to submit the work for publication.

### Author contributions

Timothy T Rogers, Conceptualization, Data curation, Formal analysis, Methodology, Visualization, Writing – original draft, Writing – review and editing; Christopher R Cox, Data curation, Formal analysis, Methodology, Writing – review and editing; Qihong Lu, Data curation, Formal analysis; Akihiro Shimotake, Data curation, Funding acquisition, Methodology, Project administration, Supervision; Takayuki Kikuchi, Takeharu Kunieda, Susumu Miyamoto, Ryosuke Takahashi, Akio Ikeda, Data curation, Methodology; Riki Matsumoto, Conceptualization, Data curation, Methodology; Matthew A Lambon Ralph, Conceptualization, Data curation, Formal analysis, Funding acquisition, Investigation, Methodology, Project administration, Supervision, Writing – original draft, Writing – review and editing

### Author ORCIDs

Timothy T Rogers 🄳 http://orcid.org/0000-0001-6304-755X
Christopher R Cox 🄳 http://orcid.org/0000-0001-8287-3865
Matthew A Lambon Ralph 🄳 http://orcid.org/0000-0001-5907-2488

### Ethics

Human subjects: All participants provided written informed consent to participate in the study and to publish the results. The study was approved by the ethics committee of the Kyoto University Graduate School of Medicine (No. C533).

### Decision letter and Author response

Decision letter https://doi.org/10.7554/eLife.66276.sa1
Author response https://doi.org/10.7554/eLife.66276.sa2

## Additional files

### Supplementary files
• Transparent reporting form

## Data availability

Code and data for running simulations, analyzing results, and replicating the ECOG decoding analysis are posted to GitHub at: https://github.com/ttrogers/DecodingDynamic, (copy archived at swh:1:rev:8cc85427c81922d107e33ef58f038fd228f042fe). This is a registered resource with RRID SCR_021099. Data for replicating all figures in the manuscript appear there.

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

# Appendix 1

## A-1: Comparison of deep model results to control models

The main paper highlights four properties of the neural decoding results observed in both the deep neural network model and in the human ECoG data: constant decodability, local temporal generalization, a widening window of generalization, and change in neural code direction in the ATL hub. We suggested that these properties arise because semantic structure is encoded as distributed activation patterns that change in highly nonlinear ways due to their situation in the deep cross-modal hub of a dynamic cortical network. This argument implies that the signature pattern would not arise in models that adopt different kinds of representation and processing mechanisms, nor in the shallower layers of the deep model. In this section we assess this implication by comparing the main results with those observed in three alternative models of semantic representation.

### Distributed versus feature-based representations

By distributed representation, we mean that many neural populations or units can jointly contribute to representation of structure even if they do not each independently encode the structure. Deep neural network models are capable of acquiring distributed representations of this kind (*Cox et al., 2014*) and may be contrasted with models proposing that semantic representations are comprised of elements that each independently detect a particular semantic feature, such as membership in a particular conceptual domain or category. We therefore considered what the decoding signature would look like in such a feature-based model. The 90 items were represented with a vector in which two elements were dedicated to each conceptual domain (animal, plant, object) and to each basic-level category (bird, fish, flower, etc; total of 9 categories). For example, a particular instance of flower would activate the two 'plant' features and the two 'flower' features; an instance of tree would activate the same two 'plant' features and two 'tree' features, etc. This yielded 24 elements total; to equate the number of features with the number of units in the deep network simulation, we added a 25th vector element that always adopted a low activation value.

We simulated the gradual activation of features over the course of processing by generating a 33-step time-series for each feature and each item presentation. All units began with an activation of 0, and features true of the stimulus would ramp up their activation according to a sigmoid function with a constant slope and a randomly sampled offset term determining when in the stimulus presentation window the feature would begin to activate. This procedure yielded a dataset analogous to the evolution of internal representations in the deep network, but with feature-based semantic representations in which features activated with randomly-sampled time-courses.

### Dynamic versus linear

By dynamic processing, we mean that units can influence themselves via feedback from the other units to which they send connections. Reciprocally connected sets of units are coupled and so behave as a dynamic system in which states evolve together over time. Often the dynamics in such a system are non-linear, producing radical changes in the ways that neural states encode information. Thus, the importance of dynamic processing in the deep neural network model can be assessed by contrasting the primary results with an alternative model that employs starting and ending representations identical to the deep model, but with intermediate states simply moving in a straight line from start to finish. We created such a model by recording the initial and final representations arising in the deep neural network model, then creating a 33-step time-series for each stimulus representing a linear interpolation of the representation moving from initial to final state. For a given stimulus, each step of the time series was created as a proportional weighted average of the initial and final states, with the first step giving all the weight to the initial representation, subsequent steps gradually shifting more weight to the final representation, and the last step giving zero weight to the initial representation. This procedure yielded a dataset in which initial and final representations were distributed identically to the deep network, but the trajectories of the representations over time were linear interpolations between these.

### Deep versus shallow

By deep network, we mean a neural network that has multiple hidden layers interposing between Visual and Verbal representation units. Depth generally allows neural networks to discover and represent more complex statistical relations amongst connected units (*Srivastava et al., 2015*).

It also allows for more complex temporal dynamics as mutual influences across distal network components take more processing time. We assessed the importance of network depth in two ways.

First we compared the behavior of the hub layer in the deep network to that of a shallow network employing just a single hidden layer containing 25 units reciprocally connected to Visual and Verbal units and parameterized identically to the hub units in the deep model. We trained the network for 30 k epochs exactly as described for the deep network, using the same training and testing patterns and procedures. This procedure yielded a dataset in which internal representations were distributed and dynamic as in the deep model but arose within a shallower network.

Second, we compared the behavior of the hub layer in the deep network to the patterns emerging across intermediate (visual hidden and verbal hidden) and shallow (visual representation and verbal representation) layers in the same model. For this comparison, we recorded the activation time-series produced in response to each visual stimulus, for every unit in the model. We then assessed the propensity for units in each layer to behave like individual feature detectors, unresponsive units, or units that appear to 'switch' their category preference over time, taking the 'switch' behavior as a marker of distributed and dynamic representation.

## Results

In the first analysis, we subjected each alternative model to the same analyses reported for the primary model and assessed whether they also show the four signature properties identified in the main paper.

*Constant decodeability.* All four models showed cross-validation accuracy reliably above chance and consistently high across the time-window once input signals reached the representation units—thus all models showed constant decodeability.

*Local temporal generalization.* **Appendix 1—figure 1A** shows a 3D MDS of the trajectories for all items through the corresponding representation space in each model. For feature-based, linear, and shallow models, the trajectories are strictly or nearly linear—only the deep, distributed and dynamic model shows the nonlinearities discussed in the main paper. Consequently the models show qualitatively different patterns of generalization over time: feature-based, linear, and shallow models show a pattern in which all classifiers generalize poorly to earlier timepoints and well to later timepoints. Thus, local temporal generalization—in which classifiers do well only for neighboring time points in both past and future—is only observed in the deep, distributed and dynamic model (A-1B).

*Widening generalization window.* In contrast to the ECoG data and the deep, distributed and dynamic model, the alternative models all show a narrowing window of temporal generalization: models fitted early in processing show good performance over a wider window than those trained later. This follows from the linear progression of internal representations observed in these models: a classification plane that succeeds for early representations must also succeed for later representations. Conversely, because representations are very well-separated toward the end of processing, classification planes that succeed later in time may not perform well for earlier representations, producing an symmetric temporal generalization profile.

*Change in code direction.* The deep, distributed and dynamic model acquired representations in which some single units, when analyzed independently, behaved like feature-detectors that change in direction over processing—with high activations initially predicting an animal stimulus, for instance, then later predicting a non-animal stimulus. A similar flipping of signal direction was also observed in the more anterior parts of the ventral temporal lobe, via the changing sign of the classifier coefficients identified in the ECoG data. We therefore considered whether a change in code direction was observed for single units considered independently in the alternative models.

Specifically, we classified each unit in each simulation as a feature-detector if its activity correlated significantly with conceptual domain in only one direction over the time-course of processing, as a switch feature if it correlated significantly in both the positive and the negative direction at different points in time, and as non-responsive if it never correlated significantly with conceptual domain. Across all five simulations and all three classification tasks, we computed the proportion of units falling into each category for each model type. The results are shown in **Appendix 1—figure 2**. For feature-based and linear models, the results are trivial, since the representations are constrained to show only non-responses or feature-like behaviors, as was

indeed observed. That such a result is observed in these cases validates the analysis. More interestingly, the shallow interactive model also learned unit responses that behaved either as feature-detectors or were non-responsive. Only the deep, distributed and dynamic model acquired units that appeared to switch their category preference.

## A-2: Stable features posteriorly with dynamic features anteriorly

Our analysis of ECoG data showed more consistent feature-like responding in more posterior ventral temporal regions, and more variability in code direction over time in vATL. We suggested that this difference between posterior and anterior regions arises because the vATL is deeper in the network and is connected cross-modally—that is, it resides in tertiary association cortex. In the model, the visual and verbal hidden layers are shallower and not directly cross-modal—they receive modality-specific inputs and interact only with the hub layer. Thus our explanation of the ECoG data has a direct model analog: if it is correct, more superficial layers should be more likely to learn static, feature-like representations, while the hub should be more prone to acquire distributed dynamic representations. We tested this hypothesis by computing the number of units in each layer that behave as feature-detectors, switch-units, or unresponsive units, taking switch-units to indicate a distributed and dynamic code.

Results are shown in *Appendix 1—figure 3*. Units in both shallow and intermediate model layers behaved like consistent feature-detectors or appeared unresponsive to the superordinate semantic category. The switching pattern diagnostic of nonlinear representational change only emerged in the deep hub layer. Note that the pattern cannot reflect overall distance in the network from the input activation, since the verbal hidden and representation units are further away from the visual inputs in the network than are the hub units but still show the same feature-like responding. Instead the pattern must reflect the centrality of the hub units in computing interactive mappings between visual and verbal representations.

Together these simulations establish that all three properties—distributed representation, dynamic processing, and network depth—conspire to yield the decoding signature observed in the ECoG data: local temporal generalization, a widening window of generalization, and neural populations whose code direction appears to change over time when considered independently.

## A-3: Independent correlation analysis of individual electrodes

The preceding simulations, together with those reported in the main paper, find model units that, considered independently, seem to 'flip' their category preference over time. A similar phenomenon was observed in the anterior part of the ventral temporal lobe when visualizing classifier weights plotted on the cortical surface: the direction and spatial organization of classifier coefficients fluctuated over time, moreso for more anterior electrodes. The coefficients show how the voltages measured at a particular location and timepoint contribute to the classifier's response when the responses of other regions are also taken into account, and so are most useful for visualizing a distributed neural code. Because coefficients do not reflect the independent correlation of an electrode's activity with the category label, however, the model and ECoG analyses are not strictly analogous. For an apples-to-apples comparison, we conducted the same independent-correlation analysis of individual electrodes in each participant. At every timepoint following stimulus onset, for each electrode across all stimuli, we computed the correlation between the electrode's measured field potential and a binary category label.

*Appendix 1—figure 4* shows these correlations plotted over time. Dotted lines indicate the correlation significance threshold of $p < 0.05$ for a two-tailed probability test on the correlation coefficient. Gray panels show electrodes that never exceeded the threshold; blue panels show electrodes that exceeded it in one direction only; red panels show electrodes that exceeded the threshold in different directions at different points in time. Two observations are warranted. First, electrodes in five of eight patients show 'switching' behavior analogous to that identified in the model, consistent with a distributed and dynamic code. Second, independent correlations never exceeded the significance threshold for one participant (patient 9)—but semantic category was nevertheless decodeable in this patient for some time windows, indicating that information can be present in a distributed code even when not discernable in individual channels.

## A-4: Decoding anterior vs posterior electrodes only

As noted in the main text, the interpretation of weights in a multivariate pattern classifier can be problematic in the presence of correlated noise (*Haufe et al., 2014*). When measurements from informative and uninformative channels are corrupted with the same noise source, a classifier will place weights of contrasting sign on both, with the weight on the uninformative channel serving to 'cancel out' the noise on the informative channel, boosting overall classifier performance. Thus an alternative explanation of the ECoG results is that semantic information is encoded solely within the stable, feature-like responses of mid/posterior regions of ventral temporal cortex, but correlated noise across channels leads the classifier to place fluctuating non-zero weights on vATL areas. Under this hypothesis, classifiers trained only on vATL electrodes should not show above-chance classification. We therefore assessed whether semantic category (animate/inanimate) can be decoded only from the more anterior electrodes.

We first computed a median split of electrodes based on their rostral/caudal location, grouping those more posterior vs anterior than –17 on the MNI y-axis. We then fit L1-regularized logistic classifiers to either the anterior electrodes only or the posterior electrodes only, using the same procedure described for the main analysis on a 100 ms moving window, computed every 50 ms across the 1650 ms post-stimulus response time. Classifiers were fit separately at each window, for each participant, and were evaluated using 10-fold cross-validation. From these data we computed, for each classifier, total number of items classified correctly across the 10 holdouts. Across participants, we then computed expected probability of correct classification and 95 % confidence intervals (from the binomial distribution) on this expectation at each window. The results are shown in *Appendix 1—figure 5*: from about 200 ms post stimulus onset, accuracy exceeded chance for both electrode subsets and remained reliably above chance across much of the decoding window. Paired t-tests on accuracy at each window showed no reliable difference between those fit to anterior versus posterior electrode sets. Thus, consistent with other arguments presented in the main paper, vATL regions encode semantic structure even considered independent of the more posterior regions that adopt a more stable and feature-like code, and both anterior and posterior regions encode some semantic structure.

## A-5: Autocorrelation analysis

In both the simulation and ECoG data, we observed that pattern classifiers generalize over a temporal window that widens over time as a stimulus is processed. We suggested this widening arises because the pattern of activation over hub units grows increasingly stable as the system generates the correct activation pattern across the rest of the network. This section considers an alternative explanation of the ECoG result: perhaps the ECoG data itself is temporally structured so that it becomes self-predictive over a broader timescale as a stimulus processed. To test this possibility, we measured the temporal autocorrelation of each electrode at lags of 1–100 ms, calculated over a 300 ms sliding window, beginning with stimulus onset and moving the window forward in 100 ms increments. If temporal autocorrelation broadens over time, we would expect significant correlation over a longer lag in windows selected from later in the time series.

*Appendix 1—figure 6* shows, for each subject, the temporal autocorrelation across lagtime, averaged across all electrodes, for 300 ms beginning at stimulus onset and sliding forward to the end of the time-series. Line colors indicate when in the time-series the 300 ms window was selected. If electrode signals grow self-predictive over a broader timespan with processing, cooler colors would show a shallower slope and a later intersection with the 0 line on the y axis. Instead all lines are practically on top of one another, and show that temporal autocorrelation within a 300 ms window drops to 0 at a lag of about 60 ms. Thus, the 'widening window' pattern observed in the ECoG temporal generalization data does not arise from shifts in the temporal autocorrelation of the data itself and must reflect an increasingly stable distributed code.

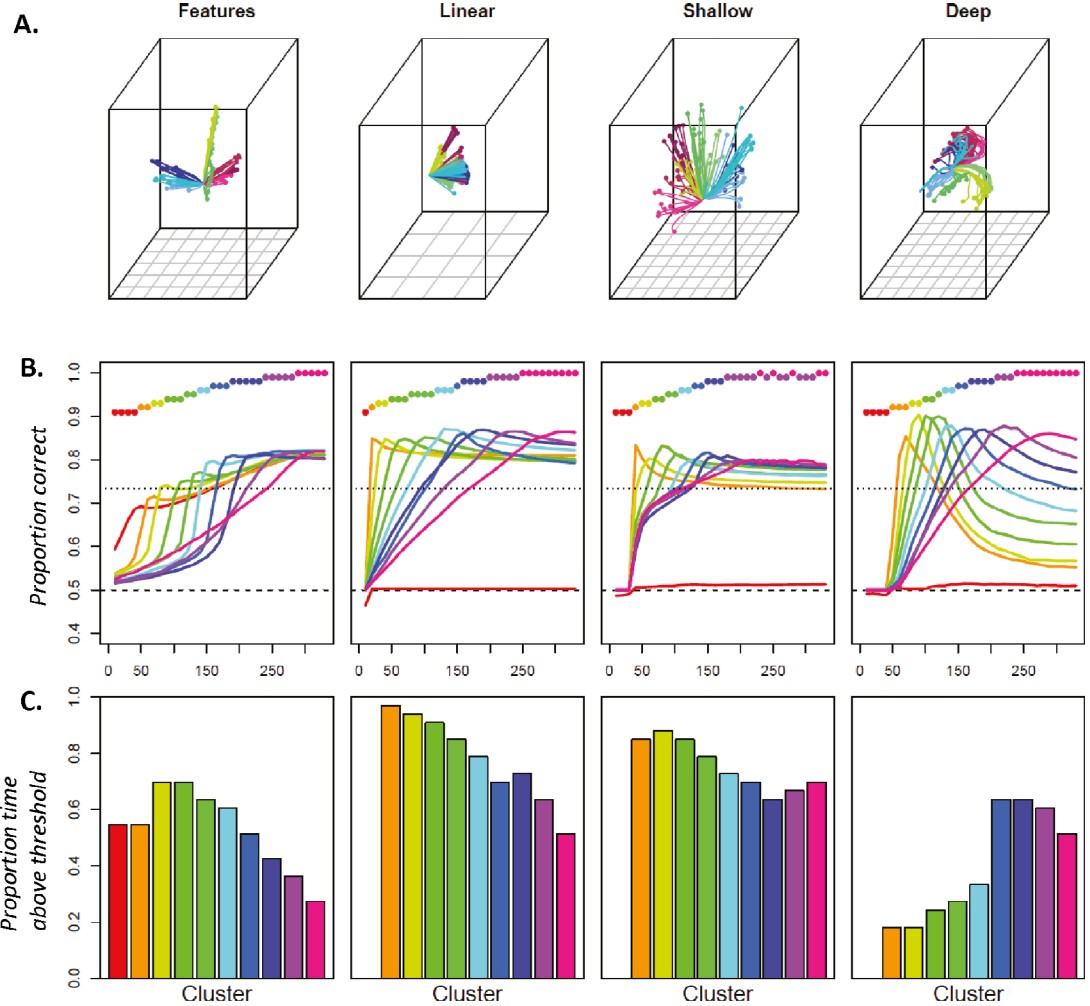

**Appendix 1—figure 1.** Comparison of simulation results for a feature-based model, a distributed linear model, a shallow recurrent network, and the deep, distributed and dynamic model. A. Multi-dimensional scaling showing the trajectory of each item through representation space under four different models. Only the deep model shows radically nonlinear change. B. Mean accuracy for clusters of classifiers under each model type. Only the deep model shows the overlapping-waves pattern. C. Proportion of time-window where classifiers in each cluster show reliably above-chance responding. Only the deep model shows a generalization window that widens over time.

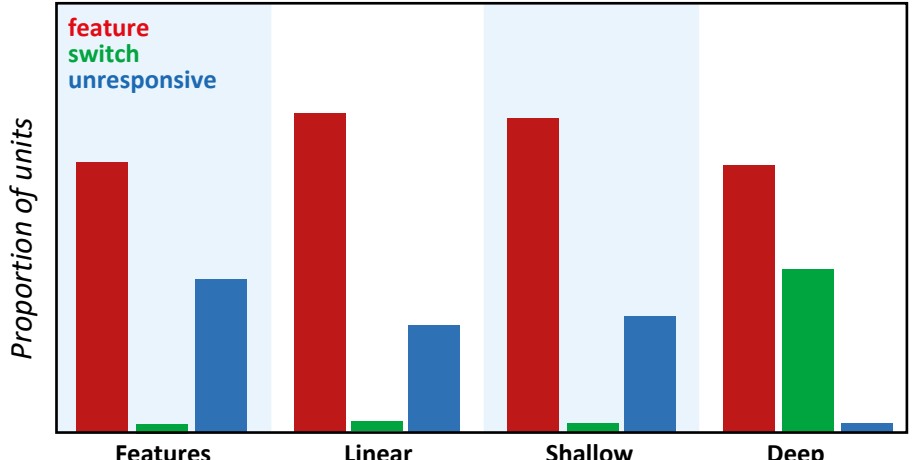

**Appendix 1—figure 2.** Types of units in each model. For each model type, the proportion of units that behave like feature-detectors (red), detectors that switch their category preference over time (green), and units that seem unresponsive to the semantic category (blue). Only the deep, distributed, dynamic model has units whose responses switch their category preference over time.

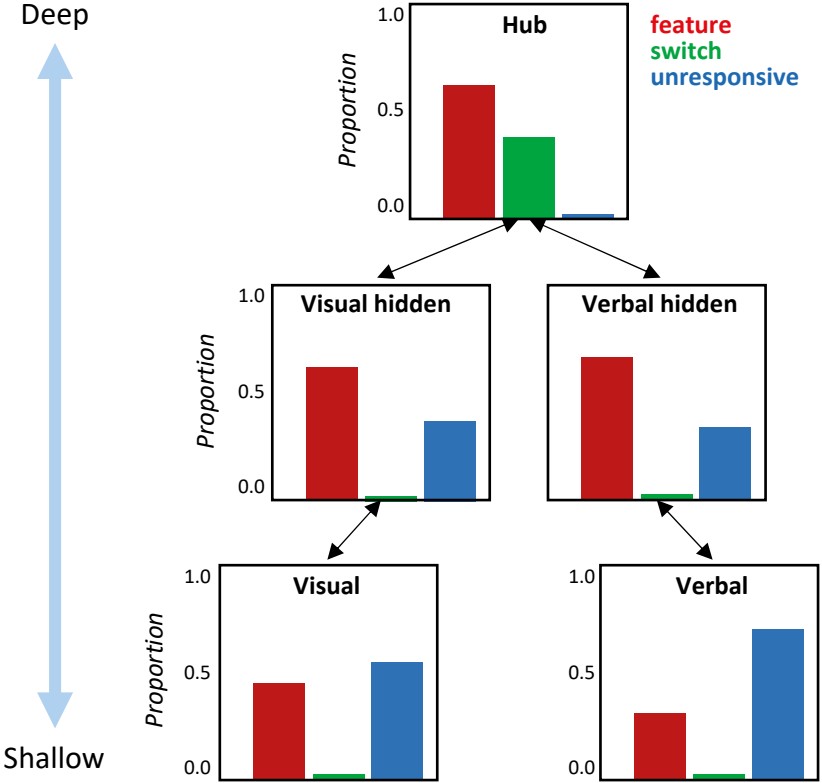

**Appendix 1—figure 3.** Types of unit in each layer of deep model. For each layer in the deep, distributed and dynamic model, the proportion of units that behave like feature-detectors (red), detectors that switch their category preference over time (green), and units that seem unresponsive to the semantic category (blue) when the model processes visual inputs. Only the hub layer of the network—the model analog to the ventral anterior temporal cortex—contained units whose responses switch their category preference over time.

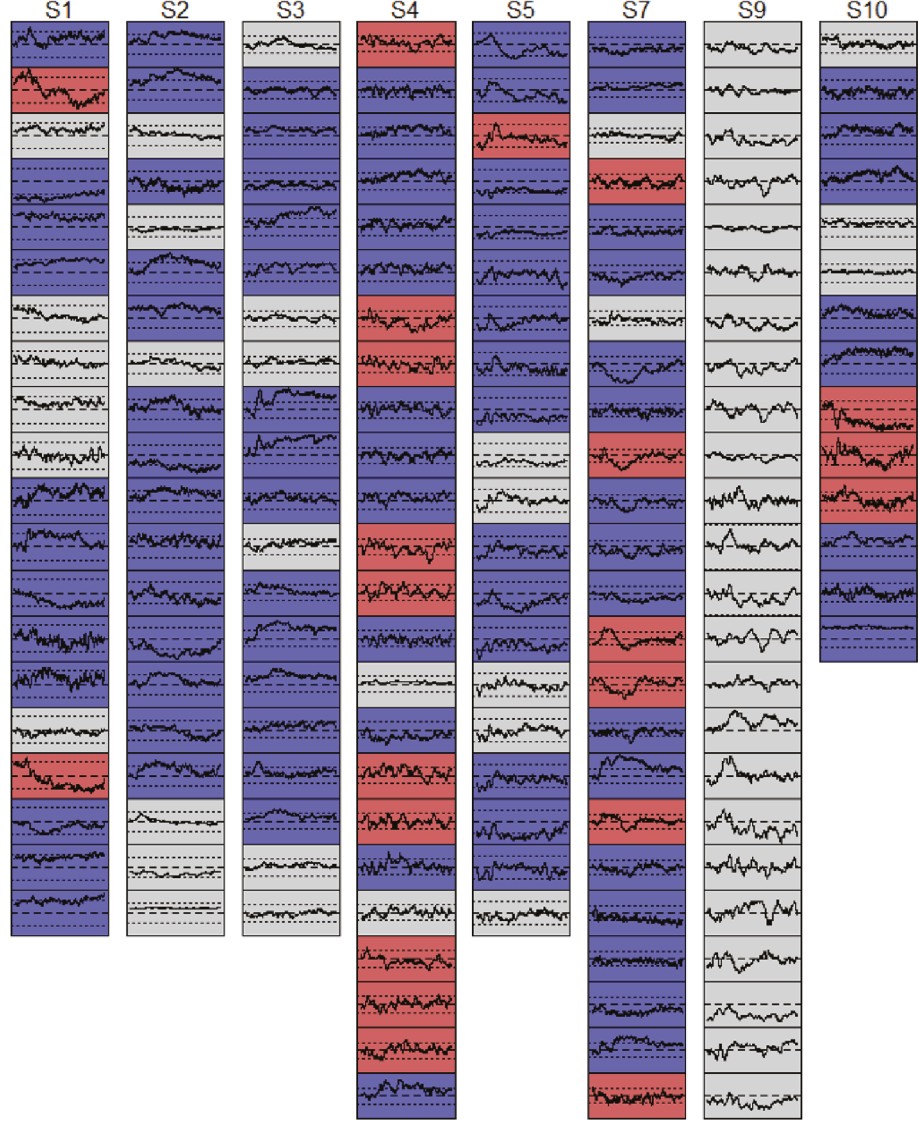

**Appendix 1—figure 4.** Independent correlations for each electrode. Each panel shows, for each electrode in each participant, the correlation over time between the measured VP for each item and the category label. Dotted lines show statistical significance thresholds for this correlations. Gray panels never exceed the threshold in either direction. Blue panels exceed it in one direction only. Red panels exceed it in both directions at different points in time.

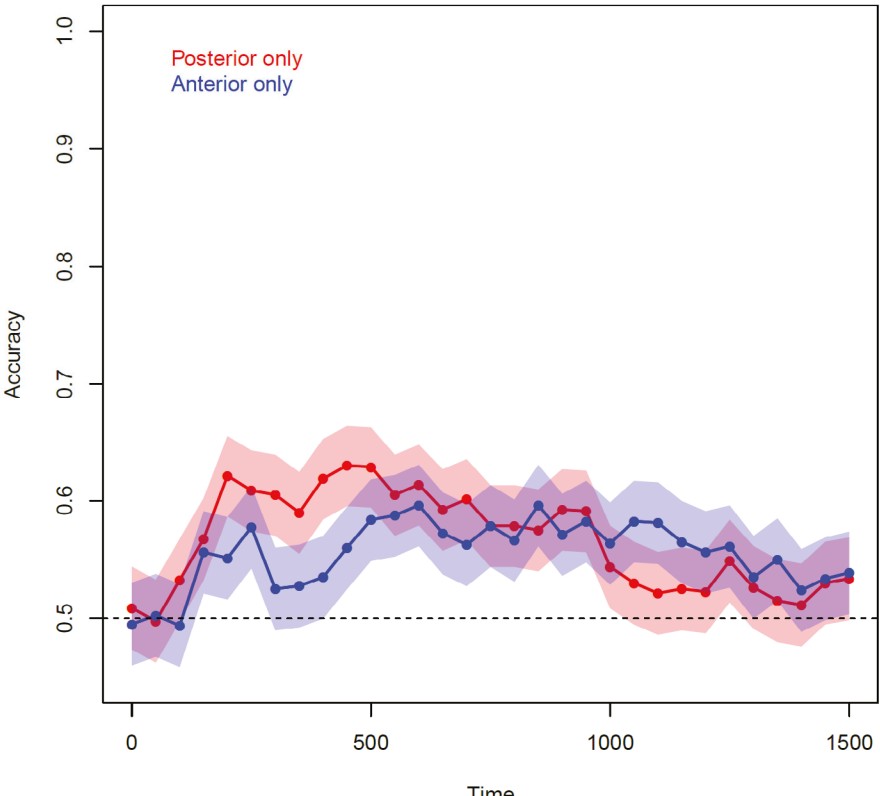

**Appendix 1—figure 5.** Accuracy for classifiers trained on anterior or posterior electrodes only. Curves show expected probability of correct classification and 95 % confidence intervals (from binomial distribution) across participants at each window for classifiers trained only on the anterior (blue) or posterior (red) half of the electrodes. Decoding accuracy exceeds chance for both subsets and does not reliably differ between these.

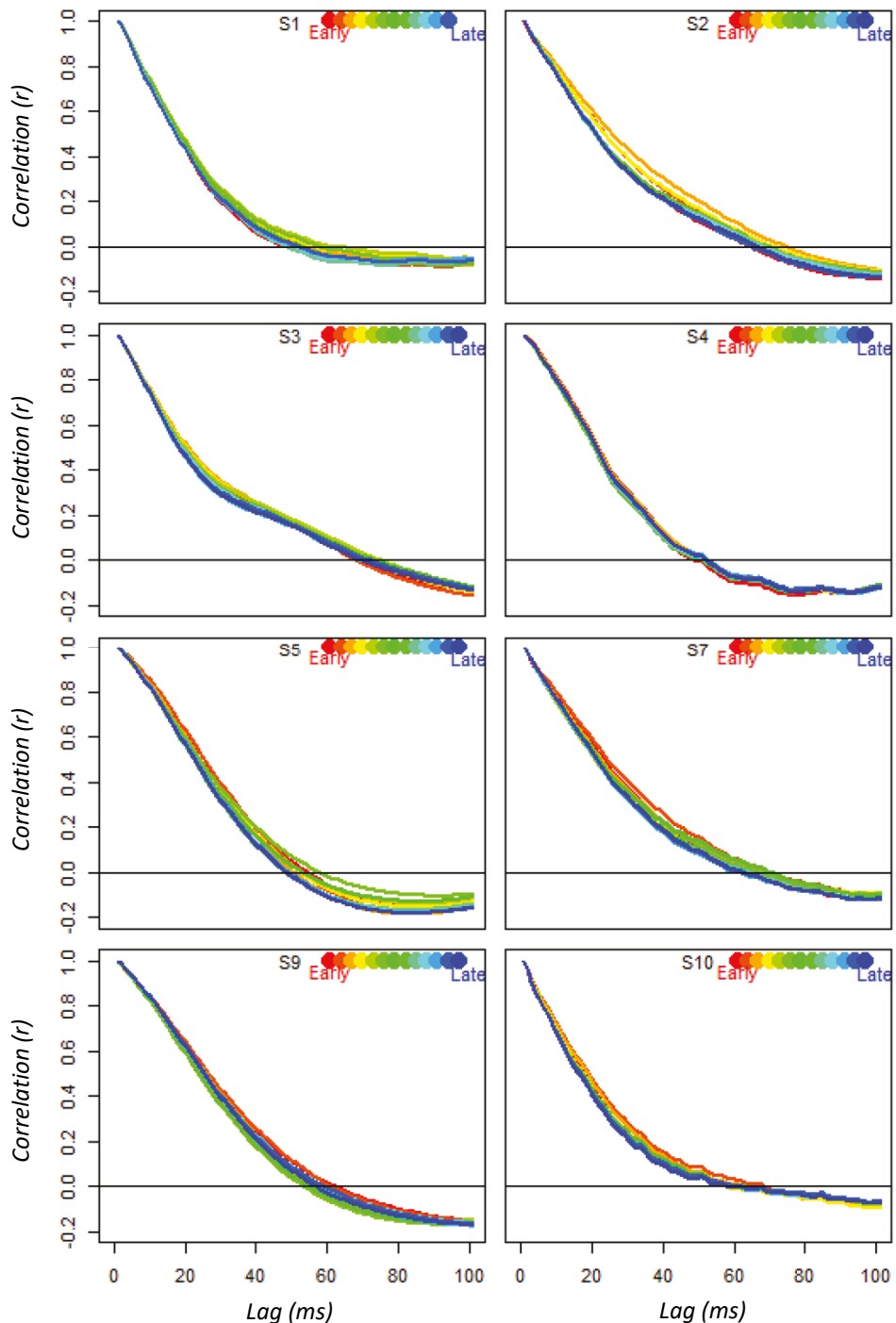

**Appendix 1—figure 6.** Temporal autocorrelation. Each panel shows the mean temporal autocorrelation curves for a 300 ms moving window, averaged over all electrodes for each subject. If VPs auto-correlate over an increasingly wide temporal window, then later curves would show a broader envelope than earlier curves. Instead the different windows sit on top of one another, showing temporal autocorrelation that decays to zero after about 60 ms.

