## [Decision Letter]

**Acceptance summary:**

This manuscript utilizes intracranial ECOG data and multivariate computational simulations and modeling to explore potential changes in semantic representations within the human anterior temporal lobe (ATL). The study addresses some long-standing issues in the collection of data from this area and the corresponding analyses. Overall, this work highlights the extremely complex and dynamic representations that can form in cortical areas that receive a wealth and diversity of inputs. This manuscript will be of interest to neuroscientists and psychologists interested in how semantic information is encoded in the brain.

**Decision letter after peer review:**

Thank you for submitting your article "Evidence for a deep, distributed and dynamic semantic code in human ventral anterior temporal cortex" for consideration by *eLife*. Your article has been reviewed by 3 peer reviewers, one of whom is a member of our Board of Reviewing Editors, and the evaluation has been overseen by Chris Baker as the Senior Editor. The following individual involved in review of your submission has agreed to reveal their identity: Stephen J Gotts (Reviewer #2).

Essential revisions:

1) Multiple reviewers noted the lack of direct quantifications/tests for some of the claims that will need to be added.

2) Some attenuation of the claims about the ATL as a semantic hub (or at least more discussion) are likely appropriate given the restricted area from which data has been collected (limiting possible direct comparisons) and the alternative interpretations brought up by Reviewer 3.

*Reviewer #1 (Recommendations for the authors):*

The main problem here lies in the interpretation of the results due to the lack of direct tests of those interpretations. If the central claim is about the distinction between anterior and posterior ATL, there must be more direct tests of that difference. In the central claim is about the need to use dynamic multivariate models to capture the differences, then the results here should be compared with simpler models.

Put the plot of the anterior electrode decoding only on the same axes as the decoding presented in the main paper. Also add in the companion analyses of the posterior electrodes alone.

*Reviewer #2 (Recommendations for the authors):*

1) Please provide more information about dynamic coding in individual patients. For example, are the red electrodes in Figure SI-4 all in the anterior aspects of the fusiform gyrus? Alternatively, can you show code switching in the vATL classifier weights for single patients (with stable coding in pATL) (as in Figure 4E) ?

2) The authors use "category correlation" in multiple figures (e.g. Figure 1B; Figure 2D; Figure SI-4), although they do not explain in much detail how this correlation is actually calculated. Please give a better verbal (or mathematical) description.

3) The current model has a lot of "compression" in terms of units per stimulus learned (25 units vs. 90 stimuli). How important is that for the code switching in the Hub hidden layer? If you were to increase all of the hidden layers to 100 units each, does this tendency decrease? As the current network tries to map similar visual inputs to dissimilar verbal outputs, it will benefit from strong attactor dynamics. It seems like the benefit will be amplified with a greater amount of compression in the hidden layers, forcing units to code 2 to 3 domains of stimuli. Is there an alternative argument as to how such compression might happen in the brain, as opposed to the network? (e.g. weight decay with effectively no limit on unit number). The context of this question is that there are many more neurons in a given brain region than the number of objects that humans know. If this variable does matter, perhaps it should be added to SI-1 ?

*Reviewer #3 (Recommendations for the authors):*

1. The methods need to be greatly improved. (1) It isn't clear to me what exactly the inputs are to the network. Is the input static (on/off) or also dynamic? This should be shown in Figure 2 (2) I found it hard to determine what was done for many things (methods). Examples: It is never explicitly stated what is decoded (living vs nonliving), what features of the LFP are used (raw voltage?)

[Editors' note: further revisions were suggested prior to acceptance, as described below.]

Thank you for submitting your article "Evidence for a deep, distributed and dynamic semantic code in human ventral anterior temporal cortex" for consideration by *eLife*. Your article has been reviewed by 2 peer reviewers, and the evaluation has been overseen by a Reviewing Editor and Chris Baker as the Senior Editor. The following individual involved in review of your submission has agreed to reveal their identity: Stephen J Gotts (Reviewer #2).

Essential revisions:

The manuscript has been improved but there are some remaining issues that need to be addressed, as outlined below:

1) There are a few outstanding considerations to be addressed in the manuscript regarding the statistical measures (see esp. Reviewer 2). These issues need to be clarified/corrected.

2) There remains a conceptual problem that needs to be addressed as well. Both reviewers have expressed some concerns with the strength of the central claim on conceptual grounds. The distinction between living and non-living is ultimately binary and a graded distinction would be stronger. If such a characterization is not possible, it must be acknowledged and the central claims attenuated.

3) There is a concern about the use of raw voltages as opposed to gamma, which can be addressed either by presenting a gamma analysis or simply attenuating the claim that motivates the already presented analysis.

*Reviewer #2 (Recommendations for the authors):*

The authors have responded well to most of my comments. However, I do have some follow-up comments on the new statistical analyses that they provide (in response to both my comments and those of the other reviewers). These comments pertain mainly to the results shown in the panels of Figure 5:

1) In panel 5A, the authors do not appear to correct for multiple comparisons (as they did in Figures 3B and 4B). Rather than the conservative Bonferroni correction used in other analyses, I think False Discovery Rate correction would be fine here (and note that it would not require non-overlapping classification windows, since positive dependence is permitted). The point of this analysis is to show local temporal generalization, so if it's still the case after correction that the best classifier performs significantly better than others in its preferred time window (which I strongly suspect will be the case), I think the point is made.

2) In panel 5B, the authors have evaluated the relationship between 'Classifier fit time' and 'Proportion of processing window reliably decoded'. However, the choice of a cut-off time for a piecewise linear fit (0-500 ms) is not independently motivated. The degrees of freedom for the correlation test are also likely to be incorrect, since the classifier windows used are overlapping (50-ms windows advanced each 10 ms). The authors should use a more automated method (rather than visual inspection) to choose the time cut-off, and they should at least use non-overlapping windows (as done in Figures 3B and 4B) to ensure that the adjacent classifier measures are statistically independent of one another.

3) In panel 5C, the authors have partly responded to one of my prior comments (asking whether or not the variable anterior coding versus stable posterior coding is reliable in individual subjects). However, there are two ambiguities present in the current analysis: i) The authors have not clarified whether each subject contributes data to each decile bin of electrodes along the y-coordinate axis, and ii) as in comment #2 above, the degrees of freedom for this test are likely incorrect. We don't know whether adjacent electrophysiological measures are statistically independent or not (and most would probably assume not). I see 2 possible solutions: i) the authors could calculate a slope of y-coordinate on 'Variance of coefficient change' for each subject and then calculate a one-sample t-test versus 0 across subjects on these slopes, or ii) the authors could instead use permutation testing on the slope as currently calculated (by randomly permuting the deciles of the y-coordinate, recalculating the slope each time for 1000 iterations or more); the actual slope could then be tested against the permuted null distribution to derive a p-value.

*Reviewer #3 (Recommendations for the authors):*

In this revised version, the authors made minor changes to the text and added statistical tests to support the key claims.

However, my major critiques the authors did not address: showing evidence of a semantic code (as opposed to the simple binary decoding) and usage of the raw voltages. Due to this I can’t see how the broad claims made are supported by this analysis.

First, while I recognize that the 'living vs non-living' distinction is used extensively in the literature for behavioral studies, I cannot see how one can conclude that a representation that allows this kind of decoding forms a 'deep semantic code'. To me these claims seem much too broad and would require much more extensive evidence to support (something like done in Huth et al. 2016, which rightly claims a semantic code). If something really is a deep semantic code, it must support encoding of many different semantic features (at different levels of abstraction, thus the deep) rather than just one. To jump from a simple binary distinction to a 'deep semantic code' is in my opinion not supported by the analysis presented here.

Second, the assertion added to the main text that raw voltages (iEEG voltages, not LFP as claimed here) are the 'closest analog to unit activity' is simply not true. There is ample evidence that gamma-band power correlates with firing rates (i.e. Nir et al. 2007) in many brain areas (which in turn correlates with the BOLD signal). But there is no evidence that firing rates correlate with ERPs in these brain areas. Indeed ERPs are largely a reflection of synaptic activity and not local unit activity. As such it is unclear what is analyzed here is reflective of processing occurring in the areas under the electrode rather than reflecting input from somewhere else.

---

## [Author Response]

Essential revisions:1) Multiple reviewers noted the lack of direct quantifications/tests for some of the claims that will need to be added.

We have added three new statistical analyses to support three central empirical claims of the paper. (The fourth claim – constant decodeability – was already statistically assessed in the original manuscript via the confidence intervals shown in Figure 4).

Specifically:

Overlapping waves: We originally noted that decoding models perform well near the windows where they are fit but decline in accuracy at further points in time, where other decoding models perform better. We provide new statistical support for this observation in an additional analysis and Figure 5A on pages 16-17. For each cluster of decoding models identified in the original group analysis, we computed a *decoding profile* separately for each participant. At each time-point we identified the best-performing decoding cluster across participants, then compared the accuracy of other decoding clusters to this best-performing cluster using within-subjects t-tests. The results showed that across subjects each cluster was reliably better than all other clusters for at least one point in time, and reliably worse than other clusters at other points in time, validating the original qualitative observation.

Widening generalization window: We originally noted that the breadth of time over which a given decoding model generalizes appears to widen over time. That is, models fit from neural signals earlier in stimulus processing generalize more narrowly than those fit to later windows. We provide a new statistical analysis and Figure 5B supporting this observation on page 17. For decoding models trained at each time-window, we computed the proportion of the full stimulus-processing window that the models decoded reliably above chance across participants. We then assessed how this breadth-of-generalization metric varied with the time at which the decoding models were fit. Breadth-of-generalization increased linearly with decoder fit-time over the first 500ms, with an r-squared of 0.8 (p < 0.00001), illustrating that the widening of the generalization window over time is statistically reliable. At 500ms breadth-of-generalization hits a ceiling where models decode above chance for almost the whole stimulus processing window.

Changing code anteriorly with stable code posteriorly*.* From plots of surface maps, we originally noted that coefficients in the decoding models appear to fluctuate substantially in magnitude and direction over time in more anterior parts of temporal cortex, and are more stable posteriorly. We have added a new statistical analysis and figure 4C supporting this observation on page 18. For each electrode in each participant, we computed how much and in what direction its coefficient in the decoding model changes from one time-window to the next non-overlapping time window. We then computed the variance of all non-zero changes for each electrode. In this variation of coefficient change metric, electrodes whose coefficients change little, or change consistently, receive small values while those whose coefficient values change in different magnitudes and directions from window to window receive high values. We divided electrodes into deciles based on their anterior/posterior MNI coordinate, and showed that variability of coefficient change increases linearly along the posterior-to-anterior axis (r-sq = 0.73, p < 0.002).

2) Some attenuation of the claims about the ATL as a semantic hub (or at least more discussion) are likely appropriate given the restricted area from which data has been collected (limiting possible direct comparisons) and the alternative interpretations brought up by Reviewer 3.

We have added some material in the general discussion (p 22) explicitly noting that, due to the limited field of view in our ECOG data, the current results do not bear on questions about the role of other cortical areas in semantic representation and processing. Reviewer 3 additionally raised the possibility that the decoding results we report reflect visual and not semantic structure. To test this possibility, we have added a new analysis showing that the living/nonliving distinction cannot be decoded from an image representation that expresses low-level visual stimulus similarity (Chamfer-matching distances). We did not consider deep-layer VGG-19 representations, since that model is trained to do semantic image classification and so does not acquire internal representations that are purely visual. We respond in more detail to this point below.

Reviewer #1 (Recommendations for the authors):The main problem here lies in the interpretation of the results due to the lack of direct tests of those interpretations. If the central claim is about the distinction between anterior and posterior ATL, there must be more direct tests of that difference. In the central claim is about the need to use dynamic multivariate models to capture the differences, then the results here should be compared with simpler models.Put the plot of the anterior electrode decoding only on the same axes as the decoding presented in the main paper. Also add in the companion analyses of the posterior electrodes alone.

We have added formal statistical tests for each core claim in our analysis of ECOG data, and hope this directly addresses the reviewer's primary concern. We have also added the requested analysis of decoding from posterior electrodes only in the Supplementary Information (p. 49), and resized the y-axis of the anterior-only plot as requested.

Reviewer #2 (Recommendations for the authors):1) Please provide more information about dynamic coding in individual patients. For example, are the red electrodes in Figure SI-4 all in the anterior aspects of the fusiform gyrus? Alternatively, can you show code switching in the vATL classifier weights for single patients (with stable coding in pATL) (as in Figure 4E) ?

We have not added plots of individual-level coefficient change over time as these are quite messy and require some statistical summary to understand. The new analysis of variation in coefficient change as a function of anterior/posterior electrode location provides such a summary and clearly shows a tight relation between these factors, which we hope addresses the reviewer's central question. Likewise the new statistical tests comparing mean accuracy of different classifier clusters over time provides evidence that the "overlapping waves" pattern is reliable across participants.

2) The authors use "category correlation" in multiple figures (e.g. Figure 1B; Figure 2D; Figure SI-4), although they do not explain in much detail how this correlation is actually calculated. Please give a better verbal (or mathematical) description.

Sorry for not being clear – this is just the correlation between a stimulus’s binary category label and the response it evokes in a unit/electrode, as now stated on page 5.

3) The current model has a lot of "compression" in terms of units per stimulus learned (25 units vs. 90 stimuli). How important is that for the code switching in the Hub hidden layer? If you were to increase all of the hidden layers to 100 units each, does this tendency decrease? As the current network tries to map similar visual inputs to dissimilar verbal outputs, it will benefit from strong attactor dynamics. It seems like the benefit will be amplified with a greater amount of compression in the hidden layers, forcing units to code 2 to 3 domains of stimuli. Is there an alternative argument as to how such compression might happen in the brain, as opposed to the network? (e.g. weight decay with effectively no limit on unit number). The context of this question is that there are many more neurons in a given brain region than the number of objects that humans know. If this variable does matter, perhaps it should be added to SI-1 ?

The relationship between number of units, compression, acquisition of structure, and relationship to real brains is a thorny one that, we think, goes somewhat beyond the scope of what we are trying to address here. For instance, the effects of increasing hidden units in the bottleneck will depend on other model parameters, such as weight decay or other forms of regularization. The question of how to measure compression in the model is also bit tricky – while there are 25 units for 90 items, 25 binary units can distinguish over 33 million distinct states, so by that metric the representation space is highly under-constrained. Rather than parametrically exploring these issues, we instead adopted a model that has been empirically validated in other ways in prior work (e.g. in the original hub-and-spokes model from Rogers et al. 2004 and the related models from Lambon Ralph, Lowe and Rogers (2007) and Chen et al. (2015)). We further note that the nonlinear dynamics in the model can't be attributed solely to compression, since the effect is greatly reduced in a model that has the same degree of compression (25 hidden units) but uses just a single shallow hidden layer rather than multiple deep layers (in simulation SI-1).

Reviewer #3 (Recommendations for the authors):1. The methods need to be greatly improved. (1) It isn't clear to me what exactly the inputs are to the network. Is the input static (on/off) or also dynamic? This should be shown in Figure 2 (2) I found it hard to determine what was done for many things (methods). Examples: It is never explicitly stated what is decoded (living vs non-living), what features of the LFP are used (raw voltage?)

As described in the Methods section on page 32, we ran the decoding analysis on LFP voltage referenced to an electrode at the scalp or galea aponeurotica, normalized to a pre-stimulus baseline, and with artifacts removed. We also note that the decoder operates on LFP voltages on pages 13, 15, 19, 20, 22, and 33. We provide extensive methodological detail in the Methods section for both the modeling and ECOG analyses. Information about inputs to the network is described on manuscript page 24 line 454 and we have clarified that the inputs are static – all dynamics arise from processing internal to the network. We aren't sure how to show this in Figure 2.2 since the input is static.

Regarding what is decoded and what features are used, the following is stated on pp 34-35:

“The pre-processed data yielded, for each electrode in each patient, a times-series of local field potentials sampled at 1000Hz over 1640ms for each of 100 stimuli. […] This procedure yielded feature vectors for all 100 items in 160 time-windows for every subject.”

[Editors' note: further revisions were suggested prior to acceptance, as described below.]

Essential revisions:The manuscript has been improved but there are some remaining issues that need to be addressed, as outlined below:1) There are a few outstanding considerations to be addressed in the manuscript regarding the statistical measures (see esp. Reviewer 2). These issues need to be clarified/corrected.

We have followed the reviewer’s recommendations closely in this regard as outlined below.

2) There remains a conceptual problem that needs to be addressed as well. Both reviewers have expressed some concerns with the strength of the central claim on conceptual grounds. The distinction between living and non-living is ultimately binary and a graded distinction would be stronger. If such a characterization is not possible, it must be acknowledged and the central claims attenuated.

We have added a new section, “Limitations and future directions,” acknowledging that we have only decoded a binary animacy distinction, outlining the challenges that face efforts to decode graded semantic similarity structure, and describing a potential avenue for doing so in future work (pp. 19-21). We also made small revisions to the title and throughout the paper to clarify that we are decoding a binary animacy distinction rather than rich and graded semantic structure. We briefly summarize these changes below under Reviewer 3’s comment.

3) There is a concern about the use of raw voltages as opposed to gamma, which can be addressed either by presenting a gamma analysis or simply attenuating the claim that motivates the already presented analysis.

In the same new section we now explicitly acknowledge issues arising from our decoding of raw voltages, explain why this has been our focus, and note how future work can begin to address these challenges. We have briefly summarized these points in the response to reviewer below.

Reviewer #2 (Recommendations for the authors):The authors have responded well to most of my comments. However, I do have some follow-up comments on the new statistical analyses that they provide (in response to both my comments and those of the other reviewers). These comments pertain mainly to the results shown in the panels of Figure 5:

Thanks for these suggestions. In the revised manuscript we have adapted our analyses as recommended.

1) In panel 5A, the authors do not appear to correct for multiple comparisons (as they did in Figures 3B and 4B). Rather than the conservative Bonferroni correction used in other analyses, I think False Discovery Rate correction would be fine here (and note that it would not require non-overlapping classification windows, since positive dependence is permitted). The point of this analysis is to show local temporal generalization, so if it's still the case after correction that the best classifier performs significantly better than others in its preferred time window (which I strongly suspect will be the case), I think the point is made.

As stated on p. 13, the figure now shows probability bands adjusted to control for FDR at a=0.05, using the p.adjust functionality in R(stats). While the figure is subtly different from the earlier plot, the central observation – each cluster performs reliably better than all other clusters at some timepoint – still holds.

2) In panel 5B, the authors have evaluated the relationship between 'Classifier fit time' and 'Proportion of processing window reliably decoded'. However, the choice of a cut-off time for a piecewise linear fit (0-500 ms) is not independently motivated. The degrees of freedom for the correlation test are also likely to be incorrect, since the classifier windows used are overlapping (50-ms windows advanced each 10 ms). The authors should use a more automated method (rather than visual inspection) to choose the time cut-off, and they should at least use non-overlapping windows (as done in Figures 3B and 4B) to ensure that the adjacent classifier measures are statistically independent of one another.

We have addressed this point by applying piecewise linear regression to only non-overlapping time windows in this data (using the *segmented* package in R). As described on pp. 14-15, we determined the number of breakpoints using an incremental approach that begins with a single linear model, then adds breakpoints and assesses improvement in model fit vs number of additional free parameters using the BIC (implemented using the *segmented* function in this package). The best model had a single breakpoint at about 473ms post stimulus onset, quite close to the point we had selected in the prior manuscript. We then fit the piecewise-linear model with one breakpoint, using only non-overlapping windows, to produce the regression lines shown in Figure 5B. As in our original analysis, the first segment shows a strong linear relationship between breadth of generalization and fit time, which accounts for 70% of the variance in the (non-overlapping) windows in this range, and is reliably non-zero at p < 0.002.

3) In panel 5C, the authors have partly responded to one of my prior comments (asking whether or not the variable anterior coding versus stable posterior coding is reliable in individual subjects). However, there are two ambiguities present in the current analysis: i) The authors have not clarified whether each subject contributes data to each decile bin of electrodes along the y-coordinate axis, and ii) as in comment #2 above, the degrees of freedom for this test are likely incorrect. We don't know whether adjacent electrophysiological measures are statistically independent or not (and most would probably assume not). I see 2 possible solutions: i) the authors could calculate a slope of y-coordinate on 'Variance of coefficient change' for each subject and then calculate a one-sample t-test versus 0 across subjects on these slopes, or ii) the authors could instead use permutation testing on the slope as currently calculated (by randomly permuting the deciles of the y-coordinate, recalculating the slope each time for 1000 iterations or more); the actual slope could then be tested against the permuted null distribution to derive a p-value.

We adopted the reviewer’s first suggestion, fitting separate linear models for each participant, then assessing whether the estimated slope is reliably positive across participants. As reported on page 16, it is reliably positive at p < 0.02, via a one-tailed t-test. We report the one-tailed result since the model predicts more variable coefficient-change in more anterior areas, though note the null-hypothesis is still rejected for a two-tailed test under conventional cutoffs.

Reviewer #3 (Recommendations for the authors):In this revised version, the authors made minor changes to the text and added statistical tests to support the key claims.However, my major critiques the authors did not address: showing evidence of a semantic code (as opposed to the simple binary decoding) and usage of the raw voltages. Due to this I can’t see how the broad claims made are supported by this analysis.First, while I recognize that the 'living vs non-living' distinction is used extensively in the literature for behavioral studies, I cannot see how one can conclude that a representation that allows this kind of decoding forms a 'deep semantic code'. To me these claims seem much too broad and would require much more extensive evidence to support (something like done in Huth et al. 2016, which rightly claims a semantic code). If something really is a deep semantic code, it must support encoding of many different semantic features (at different levels of abstraction, thus the deep) rather than just one. To jump from a simple binary distinction to a 'deep semantic code' is in my opinion not supported by the analysis presented here.

The reviewer’s comment alerted us to a potential ambiguity in the way that the word “deep” is used in different literatures. We have used the word in the manner typical of the neural network literature, referring to an architecture with multiple hidden layers interposing between input/output layers. As shown in simulations reported in the Appendix and summarized in the main paper (pp 10-11), the dynamic pattern we emphasize emerges in the deepest layer of the neural network, and does not arise within a similar model possessing only a single hidden layer. Thus the simulations suggest that the pattern we observe in the ECoG data will arise in neural networks with a deep architecture – this is the “deep” we intend when we describe a deep, distributed, and dynamic code. From this comment, it seems the reviewer may have interpreted “deep” in the classical cognitive sense introduced by Craik and Lockhart (1972), that is, as indicating a kind of semantically rich processing that goes beyond more superficial sensory or semantic processing. We have further clarified the intended meaning of “deep” in the paragraph spanning pages 10-11.

The reviewer’s initial feedback asked us to assess whether the animacy information we have emphasized was truly semantic by testing whether we could decode animacy from a measure of visual similarity. Thus, in the prior revision we added a new analysis showing that such decoding is not possible from a common unsupervised measure of similarity for binary line drawings (Chamfer matching).

The reviewer now asks for additional evidence of a richer semantic code, such as modeling more graded semantic similarity structure. We agree that decoding graded similarity structure should and will be an important focus of efforts to understand how neural systems encode semantic representations. Indeed, that project is the focus of a current international collaboration in our group. The question of how to fit a decoder to extract graded distances in a target multi-dimensional space is, however, complex with no standard solution. Unsupervised methods like correlation between neural and target RDMs don't provide such evidence, because such correlations can yield a positive result even if the signal or target truly encodes just a binary label (as was shown on the current data by Chen et al. Cortex, 2016), and can yield a negative result if a signal is present but buried in amongst many irrelevant features (as shown recently by Cox and Rogers, Journal of Neuroscience 2021). "Generative" approaches of the kind used by Mitchell, Pereira, etc, are potentially problematic because they assume the neural features are independent of one another (ie, the "meaning" of a deflection in the LFP at a given electrode is the same regardless of the states of other electrodes). This is certainly not the case in the recurrent neural network models that drive the current work. Ideally one wants a multivariate decoding model that can be fit to all data within a given time window without feature preselection (like the current paper), but which predicts the embedding of stimuli within a continuous multidimensional semantic space (instead of binary classification). Our group developed theory and analysis for such an approach in a paper targeted toward the Machine Learning community ("representational similarity learning," Oswal et al., ICML 2016), which lays out a new form of regularization for this kind of problem especially suited to neural data (the "group ordered weighted LASSO" or grOWL). In new, separate efforts we have applied it to successfully decode graded semantic structure. However, the exposition of this work requires detailed motivation and explanation of the new approach, validating it on toy data, then applying it to ECoG – steps that go substantially beyond the current paper, which focuses on how a simple neural code changes with stimulus processing.

So, following the editor’s direction, we have addressed the reviewer's concern by attenuating any unintended claims about the semantic richness of the information we are decoding. Specifically, we have added new material in the General Discussion (pp 19-20) stating that (1) the binary living/nonliving classification, while extremely common in the literature, provides limited evidence of a semantic code, since there may be other non-semantic features sufficient to differentiate these domains; (2) future work should consider whether more continuous/fine-grained aspects of semantic similarity can likewise be decoded, and if so, whether these aspects likewise change dynamically; and (3) such work will require deployment of more sophisticated decoding models capable of learning mappings from large neural datasets to a continuous low-dimensional similarity space. We have also revised the title and wording throughout the document to highlight that it is binary animacy information that is the target of our decoding.

Second, the assertion added to the main text that raw voltages (iEEG voltages, not LFP as claimed here) are the 'closest analog to unit activity' is simply not true. There is ample evidence that gamma-band power correlates with firing rates (i.e. Nir et al. 2007) in many brain areas (which in turn correlates with the BOLD signal). But there is no evidence that firing rates correlate with ERPs in these brain areas. Indeed ERPs are largely a reflection of synaptic activity and not local unit activity. As such it is unclear what is analyzed here is reflective of processing occurring in the areas under the electrode rather than reflecting input from somewhere else.

We thank the reviewer for correcting our terminology. We have replaced local field potential (LFP) with voltage potential (VP) throughout to avoid confusion.

The question of how best to relate activation in a neural network model to neurophysiological signals in the brain is also a topic of current work in our group that, we think, is somewhat more complex than the reviewer's comment suggests. Though we agree there is evidence that gamma correlates with firing rates in many brain areas, it is not clear (a) that this hold for all areas, or for the ATL specifically, (b) that power at other frequencies carries no information about local neural activity or (c) that gamma power is the only information relevant to understanding the information encoded in neural signals. Prior ECoG work has shown that object and person-naming elicits increased gamma for more perceptual areas (like posterior fusiform), but significantly alters β-band power for the anterior temporal regions that are a focus of our work (Abel et al., Journal of Neurophysiology, 2016). This may be because ATL is connected to multiple distal cortical areas since β-band oscillations may help coordinate long-distance interactions, as suggested by early simulations from Kopell et al. PNAS (2000). Likewise Tsuchiya et al. (PLOS ONE, 2008) applied multivariate decoding to ECOG signals to discriminate happy and fearful faces, finding reliable information in ventral temporal cortex in frequencies <30Hz (as well as in gamma). We are currently working on comparing the behavior of different decoder models when applied to raw voltages, power averaged at different traditional frequency bands, and decoding of the full, graded time-frequency spectrum, with an eye toward understanding what information can be extracted from each, and how time/frequency information may relate to activation patterns in ANNs. These questions, however, go beyond the focus of the current paper, which establishes that the voltages measured in vATL themselves encode stimulus information in a manner that changes over time.

So, again following the editor’s guidance, we have addressed the reviewer's concern by attenuating the strength of any conclusions resting on the suggestion that iEEG measurements only reflect local neural activity. Specifically we have (1) made revisions throughout to ensure that we do not state or imply that iEEG voltages directly measure local population firing rates, (2) added material to the general discussion explicitly noting that analysis of voltage potentials raises important questions about the extent to which the decoded information is expressed directly by firing of neural populations in vATL or by the extra-cellular effects of incoming neural signals from other areas in the brain, and (3) suggested that this question might be addressed in future work via efforts to decode time-frequency spectrograms.